# Equivariant Contrastive Learning

**Rumen Dangovski**
MIT EECS
`rumenrd@mit.edu`

**Li Jing**
Facebook AI Research
`ljng@fb.com`

**Charlotte Loh**
MIT EECS
`cloh@mit.edu`

**Seungwook Han**
MIT-IBM Watson AI Lab
`sh3264@columbia.edu`

**Akash Srivastava**
MIT-IBM Watson AI Lab
`akashsri@mit.edu`

**Brian Cheung**
MIT CSAIL & BCS
`cheungb@mit.edu`

**Pulkit Agrawal**
MIT CSAIL
`pulkitag@mit.edu`

**Marin Soljačić**
MIT Physics
`soljacic@mit.edu`

## Abstract

In state-of-the-art self-supervised learning (SSL) pre-training produces semantically good representations by encouraging them to be invariant under meaningful transformations prescribed from human knowledge. In fact, the property of invariance is a trivial instance of a broader class called equivariance, which can be intuitively understood as the property that representations transform according to the way the inputs transform. Here, we show that rather than using only invariance, pre-training that encourages non-trivial equivariance to some transformations, while maintaining invariance to other transformations, can be used to improve the semantic quality of representations. Specifically, we extend popular SSL methods to a more general framework which we name Equivariant Self-Supervised Learning (E-SSL). In E-SSL, a simple additional pre-training objective encourages equivariance by predicting the transformations applied to the input. We demonstrate E-SSL's effectiveness empirically on several popular computer vision benchmarks, e.g. improving SimCLR to 72.5% linear probe accuracy on ImageNet. Furthermore, we demonstrate usefulness of E-SSL for applications beyond computer vision; in particular, we show its utility on regression problems in photonics science. Our code, datasets and pre-trained models are available at `https://github.com/rdangovs/essl` to aid further research in E-SSL.

## 1 Introduction

Human knowledge about what makes a good representation and the abundance of unlabeled data has enabled the learning of useful representations via self-supervised learning (SSL) pretext tasks. State-of-the-art SSL methods encourage the representations not to contain information about the way the inputs are transformed, i.e. to be invariant to a set of manually chosen transformations. One such method is contrastive learning, which sets up a binary classification problem to learn invariant features. Given a set of data points (say images), different transformations of the same data point constitute positive examples, whereas transformations of other data points constitute the negatives (He et al., 2020; Chen et al., 2020). Beyond contrastive learning, many SSL methods also rely on learning representations by encouraging invariance (Grill et al., 2020; Chen & He, 2021; Caron et al., 2021; Zbontar et al., 2021). Here, we refer to such methods as Invariant-SSL (I-SSL).

The natural question in I-SSL is to what transformations should the representations be insensitive (Chen et al., 2020; Tian et al., 2020; Xiao et al., 2020). Chen et al. (2020) highlighted the importance of transformations and empirically evaluated which transformations are useful for contrastive learning (e.g., see Figure 5 in their paper). Some transformations, such as *four-fold rotations*, despite preserving semantic information, were shown to be harmful for contrastive learning. This does not mean that four-fold rotations are not useful for I-SSL at all. In fact, predicting four-fold

rotations is a good proxy task for evaluating the representations produced with contrastive learning (Reed et al., 2021). Furthermore, instead of being insensitive to rotations (invariance), training a neural network to predict them, i.e. to be *sensitive* to four-fold rotations, results in good image representations (Gidaris et al., 2018; 2019). These results indicate that the choice of making features *sensitive* or *insensitive* to a particular group of transformations can have a substantial effect on the performance of downstream tasks. However, the prior work in SSL has exclusively focused on being either entirely insensitive (Grill et al., 2020; Chen & He, 2021; Caron et al., 2021; Zbontar et al., 2021) or sensitive (Agrawal et al., 2015; Doersch et al., 2015; Zhang et al., 2016; Noroozi & Favaro, 2016; Gidaris et al., 2018) to a set of transformations. In particular, the I-SSL literature has proposed to simply remove transformations that hurt performance when applied as invariance.

To understand how sensitivity/ insensitivity to a particular transformation affects the resulting features, we ran a series of experiments summarized in Figure 1. We trained and tested a simple I-SSL baseline, SimCLR (Chen et al., 2020), on CIFAR-10 using only the *random resized cropping transformation* (solid yellow line). The test accuracy is calculated as the retrieval accuracy of a k-nearest neighbors (kNN) classifier with a memory bank consisting of the representations on the training set obtained after pre-training for 800 epochs. Next, in addition to being invariant to resized cropping, we additionally encouraged the model to be either sensitive (shown in pink) or insensitive (shown in blue) to a second transformation. We encourage insensitivity by adding the transformation to the SimCLR data augmentation, and sensitivity by predicting it (see Section 4). We varied the choice of this second transformation. We found that for some transformations, such as *horizontal flips* and *grayscale*, insensitivity results in better features, but is detrimental for transformations, such as *four-fold rotations*, *vertical flips*, *2x2 jigsaws* (4! = 24 classes), *four-fold Gaussian blurs* (4 levels of blurring) and *color inversions*. When we encourage sensitivity to these transformations, the trend is reversed. In summary, we observe that if invariance to a particular transformation hurts feature learning, then imposing sensitivity to the same transformation may improve performance. This leads us to conjecture that instead of choosing the features to be only invariant or only sensitive as done in prior work, it may be possible to learn better features by imposing invariance to certain transformations (e.g., cropping) and sensitivity to other transformations (e.g., four-fold transformations).

The concepts of sensitivity and insensitivity are both captured by the mathematical idea of equivariance (Agrawal et al., 2015; Jayaraman & Grauman, 2015; Cohen & Welling, 2016). Let $G$ be a group of transformations. For any $g \in G$ let $T_g(\boldsymbol{x})$ denote the function with which $g$ transforms an input image $\boldsymbol{x}$. For instance, if $G$ is the group of four-fold rotations then $T_g(\boldsymbol{x})$ rotates the image $\boldsymbol{x}$ by a multiple of $\pi/2$. Let $f$ be the encoder network that computes feature representation, $f(\boldsymbol{x})$. I-SSL encourages the property of "invariance to $G$," which states $f(T_g(\boldsymbol{x})) = f(\boldsymbol{x})$, i.e. the output representation, $f(\boldsymbol{x})$, does not vary with $T_g$. Equivariance, a generalization of invariance, is defined as, $\forall \boldsymbol{x} : f(T_g(\boldsymbol{x})) = T'_g(f(\boldsymbol{x}))$, where $T'_g$ is a fixed transformation (i.e., without any parameters). Intuitively, equivariance encourages the feature representation to change in a well defined manner to the transformation applied to the input. Thus, invariance is a trivial instance of equivariance, where $T'_g$ is the identity function, i.e. $T'_g(f(\boldsymbol{x})) = f(\boldsymbol{x})$. While there are many possible choices for $T'_g$ (Cohen & Welling, 2016; Bronstein et al., 2021), I-SSL uses only the trivial choice that encourages $f$ to be insensitive to $G$. In contrast, if $T'_g$ is not the identity, then $f$ will be sensitive to $G$ and we say that the "equivariance to $G$" will be non-trivial.

Therefore, in order to encourage potentially more useful equivariance properties, we generalize SSL to an Equivariant Self-Supervised Learning (E-SSL) framework. In our experiments on standard computer vision data, such as the small-scale CIFAR-10 (Torralba et al., 2008; Krizhevsky, 2009) and the large-scale ImageNet (Deng et al., 2009), we show that extending I-SSL to E-SSL by also predicting four-fold rotations improves the semantic quality of the representations. We show that this approach works for other transformations too, such as vertical flips, 2x2 jigsaws, four-fold Gaussian blurs and color inversions, but focus on four-fold rotations as the most promising improvement we obtain with initial E-SSL experiments in Figure 1.

We also note that the applications of E-SSL in this paper are task specific, meaning that the representations from E-SSL may work best for a particular downstream task that benefits from equivariance dictated by the available data. E-SSL can be further extended to applications in science; in particular, we focus on predictive tasks using (unlabelled and labelled) data collected via experiments or simulations. The downstream tasks in prediction problems in science are often fixed and can be aided by incorporating scientific insights. Here, we also explore the generality of E-SSL beyond computer vi-

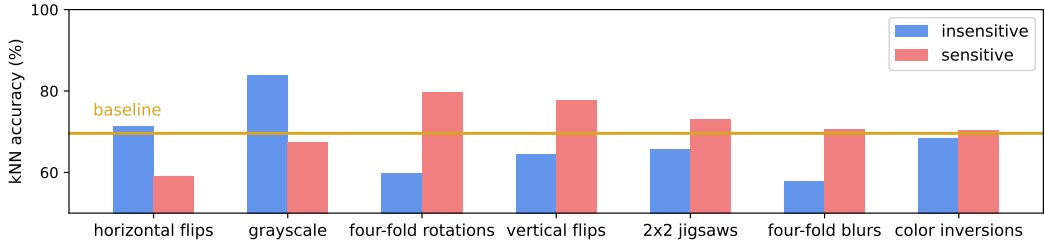

Figure 1: SSL representations should be encouraged to be either insensitive or sensitive to transformations. The baseline is SimCLR with random resized cropping only. Each transformation on the horizontal axis is combined with random resized cropping. The dataset is CIFAR-10 and the kNN accuracy is on the test set. More experimental details can be found in Section 4.

sion, on a different application: regression problems in photonics science and demonstrate examples where E-SSL is effective over I-SSL.

Our contributions can be summarized as follows:

- We introduce E-SSL, a generalization of popular SSL methods that highlights the complementary nature of invariance and equivariance. To our knowledge, we are the first to create a method that benefits from such complementarity.
- We improve state-of-the-art SSL methods on CIFAR-10 and ImageNet by encouraging equivariance to four-fold rotations. We also show that E-SSL is more general and works for many other transformations, previously unexplored in related works.
- We demonstrate the usefulness of E-SSL beyond computer vision with experiments on regression problems in photonics science. We also show that our method works both for finite and infinite groups.

The rest of the paper is organized as follows. In Section 2 we elaborate on related work. In Section 3 we introduce our experimental method for E-SSL. In Section 4 we present our main experiments in computer vision. In Section 5 provide a discussion around our work that extends our study beond computer vision. Beginning from Appendix A, we provide more details behind our findings and discuss several potential avenues of future work.

## 2 RELATED WORK

To encourage non-trivial equivariance, we observe that a simple task that predicts the synthetic transformation applied to the input, works well and improves I-SSL already; some prediction tasks create representations that can be transferred to other tasks of interest, such as classification, object detection and segmentation. While prediction tasks alone have been realized successfully before in SSL (Agrawal et al., 2015; Doersch et al., 2015; Zhang et al., 2016; Misra et al., 2016; Noroozi & Favaro, 2016; Zamir et al., 2016; Lee et al., 2017; Mundhenk et al., 2018; Gidaris et al., 2018; Zhang et al., 2019; Zhang, 2020), to our knowledge we are the first to combine simple predictive objectives of synthetic transformations with I-SSL, and successfully improve the semantic quality of representations. We found that the notion of equivariance captures the generality of our method.

To improve representations with pretext tasks, Gidaris et al. (2018) use four-fold rotations prediction as a pretext task for learning useful visual representations via a new model named RotNet. Feng et al. (2019) learn decoupled representations: one part trained with four-fold rotations prediction and another with non-parametric instance discrimination (Wu et al., 2018) and invariance to four-fold rotations. Yamaguchi et al. (2021) use a joint training objective between four-fold rotations prediction and image enhancement prediction. Xiao et al. (2020) propose to learn representations as follows: for each atomic augmentation from the contrastive learning's augmentation policy, they leave it out and project to a new space on which I-SSL encourages invariance to all augmentations, but the left-out one. The resulting representation could either be a concatenation of all projected left-out views' representations, or the representation in the shared space, before the individual projections. Our

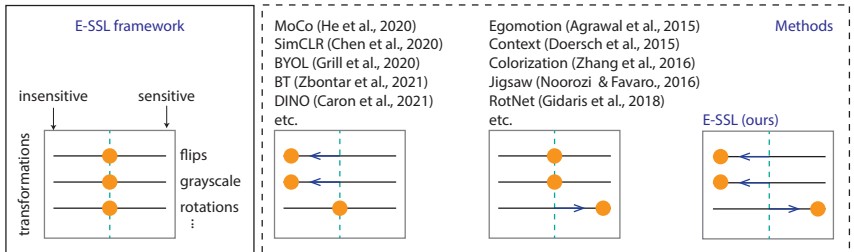

Figure 2: E-SSL framework. Left: framework. Right: methods. Egomotion, Context, Colorization and Jigsaw use other transformations than rotations, but their patterns looks like that of RotNet's. Likewise, for E-SSL can use transformations different from rotation.

method differs from the above contributions in that E-SSL is the only hybrid framework that encourages both insensitive representations for some transformations and sensitive representations for others and does not require representations to be sensitive and insensitive to a particular transformation at the same time. Thus, what distinguishes our work is the complementary nature of invariance and equivariance for multiple transformations, including finite and infinite groups.

To obtain performance gains from transformations, Tian et al. (2020) study which transformations are the best for contrastive learning through the lens of mutual information. Reed et al. (2021) use four-fold rotations prediction as an evaluation measure to tune optimal augmentations for contrastive learning. Wang & Qi (2021) use strong augmentations to improve contrastive learning by matching the distributions of strongly and weakly augmented views' representation similarities to a memory bank. Wang et al. (2021) provide an effective way to bridge transformation-insensitive and transformation-sensitive approaches in self-superived learning methods via residual relaxation. A growing body of work encourages invariance to domain agnostic transformations (Tamkin et al., 2021; Lee et al., 2021; Verma et al., 2021) or strengthens invariance with regularization (Foster et al., 2021). Our framework is different from the above works, because we work with transformations that encourage equivariance beyond invariance.

To understand and improve equivariant properties of neural networks, Lenc & Vedaldi (2015) study emerging equivariant properties of neural networks and (Cohen & Welling, 2016; Bronstein et al., 2021) construct equivariant neural networks. In contrast, our work does not enforce strict equivariance, but only encourages equivariant properties for the encoder network through the choice of the loss function. While strict equivariance is concerned with groups, some of the transformations, such as random resized cropping and Gaussian blurs, may not even be groups, but they could still be analyzed in the E-SSL framework. Thus, ours is a flexible framework, which allows us to consider a variety of transformations and how the encoder might exhibit equivariant properties to them.

## 3 METHOD

Our method is designed to test our primary conjecture that a *hybrid approach* of sensitive and insensitive representations learns better features. Surprisingly, this hybrid approach is not yet present in SSL, as Figure 2 illustrates. In this figure, we can view transformations in SSL as "levers." Each downstream task has an optimal configuration of the levers, which should be tuned in the SSL objective: left for insensitive and right for sensitive representations. E.g., make representations insensitive to horizontal flips and grayscale and sensitive to four-fold rotations, vertical flips, 2x2 jigsaws, Gaussian blurs or color inversions. Formally, insensitive and sensitive features correspond to trivial and regular group representations, respectively. Here, we present an effective method to achieve this control.

Let $f(\cdot; \boldsymbol{\theta}_f)$ with trainable parameters $\boldsymbol{\theta}_f$ be a backbone encoder. Analogously, let $p_1(\cdot; \boldsymbol{\theta}_{p_1})$ be a projector network for the I-SSL loss. There might be an extra prediction head and parameters, depending on the objective, which we omit for similicity. Let $p_2(\cdot; \boldsymbol{\theta}_{p_2})$ be the predictor network for encouraging sensitivity, which we will call "predictor for equivariance." We share the backbone encoder $f$ jointly for I-SSL and the objective of predicting the transformations from the backbone representations. Let $\ell_{\text{I-SSL}}$ be the I-SSL loss and $\ell_{\text{E-SSL}}$ be the added E-SSL loss that encourages

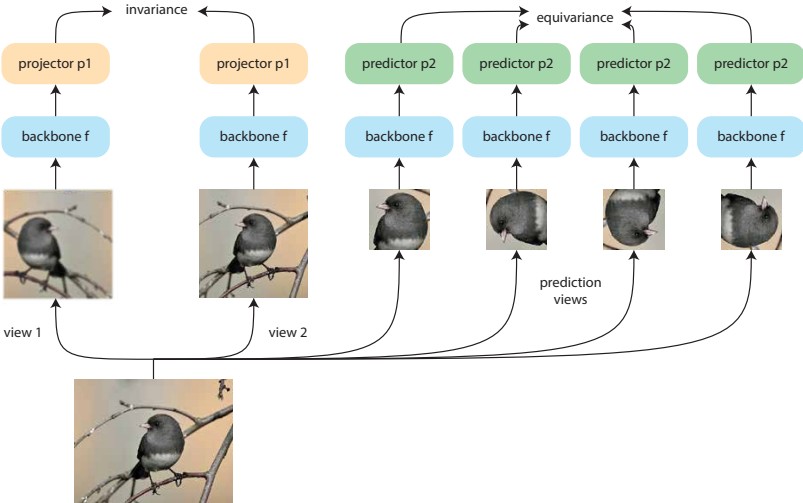

Figure 3: Sketch of E-SSL with four-fold rotations prediction, resulting in a backbone that is sensitive to rotations and insensitive to flips and blurring. ImageNet example n01534433:169.

sensitivity to a particular transformation. Let the parameter $\lambda$ be the strength of the E-SSL loss. The optimization objective for an image $\boldsymbol{x}$ with views $\{\boldsymbol{x}'\}$ in the batch is given as follows

$$\underset{\boldsymbol{\theta}_f, \boldsymbol{\theta}_{p_1}, \boldsymbol{\theta}_{p_2}}{\arg\min}\; \ell_{\mathrm{SSL}}(p_1(f(\{\boldsymbol{x}'\}; \boldsymbol{\theta}_f); \boldsymbol{\theta}_{p_1})) + \lambda \mathbb{E}_{g \in G}\left[\mathrm{PredictionLoss}(g, p_2(f(T_g(\boldsymbol{x}'); \boldsymbol{\theta}_f); \boldsymbol{\theta}_{p_2})\right] \quad (1)$$

where $\ell_{\mathrm{E\text{-}SSL}}$ (the expectation in the second summand) can take either one or all of the views, but we take only one for simplicity. The goal of $\ell_{\mathrm{E\text{-}SSL}}$ is to predict $g$ from the representation $p_2(f(T_g(\boldsymbol{x}'); \boldsymbol{\theta}_f); \boldsymbol{\theta}_{p_2})$, which encourages equivariance to the group of transformations $G$. The PredictionLoss could be either a cross entropy loss for finite groups or L1/ MSE loss for infinite groups. In practice we replace the expectation with an unbiased estimate. Most of our experiments in this paper focus on finite groups, but we show one example for an infinite group in Appendix F.

E-SSL can be constructed for any semantically meaningful transformation (see for example, Figure 1). From Figure 1 we choose four-fold rotations as the most promising transformation and we fix it for the upcoming section. As a minor motivation, we also present empirical results about the similarities between four-fold rotations prediction and I-SSL in Appendix C. In particular, both tasks benefit from the same data augmentation. Figure 3 sketches how our construction works for predicting four-fold rotations. In particular, we sample each of the 4 possible rotations uniformly and use the cross entropy loss for the PredictionLoss in Equation 1.

**What transformations could work for E-SSL?** A common property of the successful transformations we have studied up to this point is that they form groups in the mathematical sense, i.e. (*i*) each transformation is invertible, (*ii*) composition of two transformations is part of the set of transformations and (*iii*) compositions are associative.

In this paper, we encourage equivariance to a group of transformation by predicting them. This does not guarantee that the encoder we learn will be strictly equivariant to the group. In practice we observe that invariance and equivariance is well encouraged by the training objectives we use (see Appendix D.3 for detailed analysis). In fact, even strict equivariance is possible, i.e. there exists an encoder that is non-trivially equivariant, under a reasonable assumption which is formulated as follows. Let $X$ be the set of all images. Let $G$ be a group whose elements $g \in G$ transform $X$ via the function $T_g\colon X \to X$. Let $X' = \{T_g(\boldsymbol{x}) \mid g \in G, \boldsymbol{x} \in X\}$ be the set of all transformed images. Let $f(\cdot; \boldsymbol{\theta})\colon X' \to S$ be an encoder network that we learn with parameters $\boldsymbol{\theta}$. We write $f(\cdot) \equiv f(\cdot; \boldsymbol{\theta})$ for simplicity. Finally, let $S = \{f(\boldsymbol{x}') \mid \boldsymbol{x}' \in X\}$ be the set of all representations of the images in $X'$. The following is our statement.

**Proposition 1** (Non-trivial Equivariance). *Given $T_g\colon X' \to X'$ for the group $G$, there exists an encoder $f\colon X' \to S$ that is non-trivially equivariant to the group $G$ under the assumption that if $f(T_g(\boldsymbol{x})) = f(T_{g'}(\boldsymbol{x}'))$ then $g = g'$ and $\boldsymbol{x} = \boldsymbol{x}'$ for all $g, g' \in G$ and $\boldsymbol{x}, \boldsymbol{x}' \in X$.*

We defer the proof to Appendix B. The significance of this proof is that it explicitly constructs a non-trivially equivariant encoder network for groups $G$ if the assumption is satisfied. The intuition of the assumption is that if the representations of two transformed inputs are the same, the inputs should coincide, and likewise the transformations. More formally, this assumption reflects the condition when the dataset contains only one element of each group orbit. We speculate that satisfying this assumption is reasonable for the datasets in this work, since we observe a natural setting of the data, e.g. horizontal mirror symmetry in Gpm, and we consider transformations that disturb this natural setting. In Appendix G we also show that E-SSL is crucial for the Flowers-102 dataset for which this assumption might be less clear. In Appendix H we also present a natural modification of E-SSL for scenarios, where that assumption is violated.

Could other transformations still help? To motivate our work, in Figure 1 we observed additional transformations that could be useful, such as vertical flips, 2x2 jigsaws, four-fold Gaussian blurs and color inversions. All of these transformations are groups, except for four-fold Gaussian blurs. Each element of Gaussian blurs is invertible (de-blurring), but the inverse is not a transformation in the set. Interestingly, we observe that four-fold Gaussian blurs still improve the baseline, which means the success of E-SSL may not be limited to groups.

We might also consider combining the prediction of multiple transformations to encourage sensitivity to all of them. However, the gains we saw in Figure 1 may not add up when we combine transformations, because they may not be independent. The gains may also depend on the transformations that we choose for I-SSL. While we see combinations of transformations as promising future work, we focus on a single transformation to make a clear presentation of E-SSL.

## 4 EXPERIMENTS

### 4.1 SETUPS

**CIFAR-10 setup.** We use the CIFAR-10 experimental setup from (Chen & He, 2021). We consider two simple I-SSL methods: SimCLR (with InfoNCE loss (Oord et al., 2018) and temperature 0.5) and SimSiam (Chen & He, 2021). We were able to obtain baseline results close to those in (Chen & He, 2021). The predictor for equivariance takes a smaller crop with size 16x16. We report performance on the standard linear probe. We tune $\lambda$ to 0.4 both for SimCLR and SimSiam (full tuning in Table 6 in Appendix D). Remaining experimental details can be found in Appendix D.

**ImageNet setup.** We use the original augmentation setting for each method. The predictor for equivariance takes a smaller crop with size 96x96. We use a ResNet-50 (He et al., 2016) backbone for each method. In terms of optimizer and batch size settings, we follow the standard training recipe for each method. For our SimCLR experiments we use a slightly more optimal implementation that uses BYOL's augmentations (i.e. it includes *solarization*), initializes the ResNet with zero BatchNorm weights and uses the InfoNCE loss with temperature 0.2.

**Photonic-crystals setup.** Photonic crystals (PhC) are periodically-structured materials engineered for wide ranging applications by manipulating light waves (Yablonovitch, 1987; Joannopoulos et al., 2008). The density-of-states (DOS) is often used as a design metric to engineer the desired properties of these crystals and thus here, we consider the regression task of predicting the DOS of PhCs. Examples of this dataset are depicted in Section 5 and further details can be found in Appendix F. The use of symmetry or invariance knowledge is common in scientific problems; here, the DOS labels are invariant to several physical transformations of the unit cell, namely, rolling translations (due to its periodicity), operations arising from the symmetry group ($C_{4v}$) of the square lattice, i.e. rotations and mirror flips, and refractive scaling. We construct an encoder network comprising of simple convolutional and fully-connected layers (see Appendix F) and create various synthetic datasets to investigate encouragement of equivariance. After SSL/ E-SSL, we fine-tune the network with L1 loss; for better interpretability of prediction accuracies, we use a relative error metric (Liu et al., 2018; Loh et al., 2021) for evaluation, given by $\ell_{\text{DOS}} = (\sum_\omega |\text{DOS}^{\text{pred}} - \text{DOS}|)/(\sum_\omega \text{DOS})$, reported in (%). We defer the results to Section 5, because of the novelty of the experimental setup.

**The predictor $p_2$ for E-SSL.** The predictor is a 2 layer MLP for CIFAR-10 and Photonic-crystals, and a 3 layer MLP for ImageNet, followed by a linear head that produces the logits for the an n-way

---

**Algorithm 1** PyTorch-style pseudocode for E-SSL, predicting four-fold rotations.

```
# f: backbone encoder network
# p1: projector network for I-SSL
# p2: predictor network for E-SSL
# ssl_loss: loss function for I-SSL
# lambda: weight of the E-SSL

for x in loader:
    # large views for SSL and small view for EE
    V_large = augment(x, small_crop=False) # list of views
    v_small = augment(x, small_crop=True) # change: crop with size=96 and scale=(0.05, 0.14)

    # loss
    loss_invariance = ssl_loss(p1(f(V_large)))
    labels = [0] * N + [1] * N + [2] * N + [3] * N # 4Nx1
    v_cat = cat([v_small] * 4, dim=0) # 4Nx3x96x96
    v_equivariance = rot90(v_cat, labels) # constructing the rotated views

    logits = p2(f(v_equivariance)) # 4Nx4
    loss_equivariance = CrossEntropyLoss(logits, labels) # rotation prediction
    loss = loss_invariance + lambda * loss_equivariance

    # optimization step
    loss.backward()
    optimizer.step()
```

---

Table 1: Linear probe accuracy (%) on CIFAR-10. Models are pre-trained for 800 epochs. Baseline results are from Appendix D in (Chen & He, 2021). Standard deviations are from 5 different random initializations for the linear head. Deviations are small because the linear probe is robust to the seed.

| Method | SimCLR (Chen et al., 2020) | SimSiam (Chen & He, 2021) |
|---|---|---|
| Baseline (Chen & He, 2021) | 91.1 | 91.8 |
| Baseline (our reproduction) | $92.0_{\pm 0.0}$ | $91.6_{\pm 0.0}$ |
| E-SSL (ours) | $\mathbf{94.1}_{\pm 0.0}$ | $\mathbf{94.2}_{\pm 0.1}$ |
| Ablating E-SSL | | |
| Single random rotation | $93.4_{\pm 0.0} (\downarrow 0.7)$ | $92.6_{\pm 0.0} (\downarrow 1.6)$ |
| Linear predictor for equivariance | $93.3_{\pm 0.0} (\downarrow 0.8)$ | $93.4_{\pm 0.0} (\downarrow 0.8)$ |
| No SSL augmentation in equivariance views | $92.7_{\pm 0.1} (\downarrow 1.4)$ | $92.0_{\pm 0.1} (\downarrow 2.2)$ |
| Alternatives to E-SSL | | |
| Disentangled representations | $91.3_{\pm 0.0} (\downarrow 2.7)$ | $91.1_{\pm 0.0} (\downarrow 3.1)$ |
| Insensitive instead of sensitive | $86.3_{\pm 0.1} (\downarrow 7.8)$ | $86.1_{\pm 0.1} (\downarrow 8.1)$ |

classification (for example four-fold rotations is 4-way classification), or a single node for the continuous group experiment. The predictor's hidden dimension is shared across all layers and it equals 2048 for CIFAR-10 and ImageNet and 512 for PhC. After each linear layer, there is a Layer Normalization (Ba et al., 2016) followed by ReLU. We experimented with Batch Normalization (Ioffe & Szegedy, 2015) (with trainable affine parameters) instead of Layer Normalization, but did not observe any significant changes. For some experiments, we discovered that removing the last ReLU from the MLP improves the results slightly. In particular, for SimSiam on CIFAR-10 and for all models on ImageNet we omit the last ReLU.

Finally, Algorithm 1 presents pseudocode for E-SSL with four-fold rotations on ImageNet. In our implementation, we use smaller resolution for the rotated images, so that we can fit all views on the same batch and have minimal overhead for pre-training (additional details in Table 9 in Appendix E).

## 4.2 Main results

**CIFAR-10 results.** To highlight the benefits of our method, Table 1 demonstrates the improvement we obtain by using E-SSL on top of SimCLR and SimSiam and then shows different ablations and alternative methods. We label the E-SSL extensions as E-SimCLR and E-SimSiam respectively. We

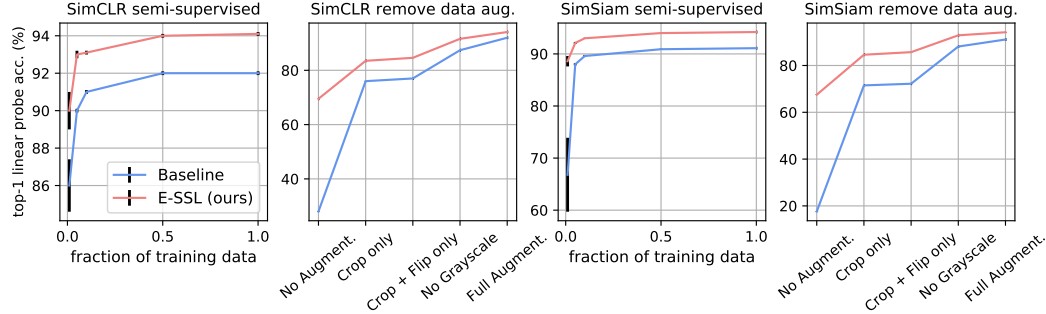

Figure 4: Reducing the labels for training and the data augmentation for pre-training on CIFAR-10. Error bars for 5 different training data splits.

Table 2: Linear probe accuracy (%) on ImageNet. Each model is pre-trained for 100 epochs. Baseline results are from Table B.1 in (Chen et al., 2020) from Table 4 in (Chen & He, 2021). Numbers marked with * use a less optimal setting than our reproduction for SimCLR (see ImageNet setup).

| Method | SimCLR (Chen et al., 2020) | SimSiam (Chen & He, 2021) | Barlow Twins (Zbontar et al., 2021) |
|---|---|---|---|
| Baseline (Chen et al., 2020) | 64.7* | - | - |
| Baseline (Chen & He, 2021) | 66.5* | 68.1 | - |
| Baseline (our reproduction) | 67.3 | 68.1 | 66.9 |
| E-SSL (ours) | **68.3** | **68.6** | **68.2** |

observe that we can increase a tuned baseline accuracy by about $2 - 3\%$. When ablating E-SSL, we see that each component of E-SSL is important. Most useful is the SSL augmentation applied on top of the rotated views. We also study alternatives to E-SSL. With "Disentangled representations" we investigate whether a "middle ground" is optimal for E-SSL: half of the representation to be insensitive to a transformation and the other half to be sensitive to the same transformation. This results in degradation of performance, which reflects our hypothesis that the representations should be either insensitive or sensitive. We conducted this experiment by using four-fold rotations in I-SSL for half of the representation and E-SSL for the other half. Finally, making the representations "Insensitive instead of sensitive" to four-fold-rotations hurts the performance significantly, as it is also observed in Figure 1, and in (Chen et al., 2020; Xiao et al., 2020).

Figure 4 reveals that E-SSL is more robust to removing transformations for I-SSL or reducing the labels for training. For example, E-SimCLR and E-SimSiam with only random resized cropping obtain 83.5% and 84.6% accuracies. Encouraging sensitivity to one transformation, namely four-fold-rotations, can reduce the need for selecting many transformations for I-SSL and with only 1% of the training data, E-SimCLR and E-SimSiam achieve 90.0± 1.0% and 88.6± 1.0% respectively.

**ImageNet results.** Table 2 demonstrates our main results on the linear probe on ImageNet after pre-training with various state-of-the-art I-SSL methods and their E-SSL versions. By only sweeping $\lambda$ and slightly reducing the original learning rate for SimSiam we obtain consistent 1%/ 0.5%/ 1.3% improvements for SimCLR/ SimSiam/ Barlow Twins respectively. Additionally, in Table 3 we observe consistent benefits of using E-SSL with longer pre-training. Finally, after 800 epochs of pre-training E-SimCLR achieves **72.5%**, which is 0.6% better than SimCLR's 71.9% baseline.

Table 3: Linear probe accuracy (%) on ImageNet with longer pre-training. "BT" is short for "Barlow Twins."

| Method | pre-training epochs | | |
|---|---|---|---|
| | 100 | 200 | 300 |
| SimCLR (repro) | 67.3 | 69.7 | 70.6 |
| E-SimCLR (ours) | **68.3** | **70.5** | **71.5** |
| BT (repro) | 66.9 | 70.0 | 71.1 |
| E-BT (ours) | **68.2** | **71.0** | **71.9** |

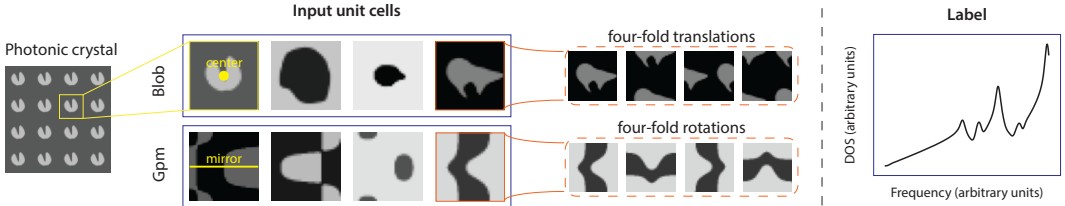

Figure 5: PhC datasets with transformations for sensitivity. The regression task is to predict the DOS labels (an example of a label in $\mathbb{R}^{400}$ is shown on the right) from 2D square periodic unit cells (examples of the inputs in $\mathbb{R}^{32\times32}$ are shown on the left). We consider two types of input unit cells; at the top is the Blob dataset where the feature variation is always centered; at the bottom is the Group pm (Gpm) dataset where inputs have a horizontal mirror symmetry.

Table 4: Fine-tuning the backbone on PhC datasets using 3000/ 2000 labelled train/ test samples. Relative error (%) is $\ell_{\mathrm{DOS}} = (\sum_\omega |\mathrm{DOS}^{\mathrm{pred}} - \mathrm{DOS}|)/(\sum_\omega \mathrm{DOS})$. Lower is better. SimCLR for Blob includes $C_{4v}$ (rotations and flips); SimCLR for Gpm includes rolling translations and mirrors. E-SimCLR encourages the features to be sensitive to the selected transformation explained in the text (four-fold translations for Blob and four-fold rotations for Gpm). "+ Transform" means adding this transformation to SimCLR. Error bars are for 3 different training data splits.

| PhC Dataset | Supervised | SimCLR | SimCLR + Transform | E-SimCLR (ours) |
|---|---|---|---|---|
| Blob | $1.068_{\pm 0.015}$ | $0.987_{\pm 0.005}$ | $0.999_{\pm 0.005}$ | $\mathbf{0.974}_{\pm 0.009}$ |
| Gpm | $3.212_{\pm 0.041}$ | $3.122_{\pm 0.002}$ | $3.139_{\pm 0.005}$ | $\mathbf{3.091}_{\pm 0.006}$ |

## 5 DISCUSSION

To show that other domains benefit from E-SSL in a qualitatively similar way to the applications in the previous section, here we introduce two datasets in photonics science. Figure 5 depicts the datasets, i.e. input-label pairs consisting of 2D square periodic unit cells of PhCs and their associated DOS. The physics of the problem dictates that the DOS is invariant to (rolling) translations, scaling of all pixels by a fixed positive factor, and operations of the $C_{4v}$ symmetry group, i.e. rotations and mirror flips. In choosing the transformations that E-SSL should encourage sensitivity to, we observe that the transformations that have worked for CIFAR-10 and ImageNet disturb the natural setting of the data (e.g. rotations disturb the natural upright setting of images). Thus, we encourage sensitivity to transformations that fit this observation, and insensitivity to the rest of the transformations.

In Figure 5, the top dataset is a "Blob" dataset where the shape variation in each image is centered. We encourage sensitiviy to the group of *four-fold translations*, given by $G = \{e, h, v, hv\}$, where $h$ and $v$ are 1/2-unit cell translations in the horizontal and vertical axis, respectively, $e$ is the unit element (no transformation) and $hv$ is the composition of $h$ and $v$. In the bottom dataset of Figure 5, the PhC unit cells are generated to have a horizontal mirror symmetry, i.e. we use the 2D wallpaper (or crystallographic plane) group **pm**. We encourage sensitivity to the group of *four-fold rotations* (the same group we used for CIFAR-10 and ImageNet), since rotating any of the images disturbs the (horizontal) mirror symmetry. More accurately, since only $\pm\pi/2$ rotations disturb the symmetry, we separate them in two classes, $\{\pi/2, -\pi/2\}$ and $\{0, \pi\}$, and perform binary prediction in E-SSL.

Table 4 shows the results of fine-tuning the backbone and an additional DOS-predictor head (see Appendix F) with 3000 labelled samples for this regression task. We observe that encouraging sensitivity to the selected transformations (via E-SimCLR) leads to the largest reduction in the error. On the contrary, including these transformations to SimCLR (indicated by "+ Transform") increases the error. Furthermore, we explore scaling transformations and show that E-SSL can be generalized to infinite groups (see Appendix F). This supports our observations about the usefulness of E-SSL over I-SSL and demonstrates E-SSL's generality beyond computer vision.

## ACKNOWLEDGEMENTS

We thank Shurong Lin, Kristian Georgiev, Alexander Atanasov, Peter Lu and the anonymous reviewers for fruitful conversations and support. R.D. dedicates this work to the memory of Boyko Dangovski.

The authors acknowledge the MIT SuperCloud and Lincoln Laboratory Supercomputing Center (Reuther et al., 2018) for providing HPC and consultation resources that have contributed to the research results reported within this paper/report.

Research was sponsored by the United States Air Force Research Laboratory and the United States Air Force Artificial Intelligence Accelerator and was accomplished under Cooperative Agreement Number FA8750-19-2-1000. The views and conclusions contained in this document are those of the authors and should not be interpreted as representing the official policies, either expressed or implied, of the United States Air Force or the U.S. Government. The U.S. Government is authorized to reproduce and distribute reprints for Government purposes notwithstanding any copyright notation herein.

This material is also based in part upon work supported by the Air Force Office of Scientific Research under the award number FA9550-21-1-0317 and the U. S. Army Research Office through the Institute for Soldier Nanotechnologies at MIT, under Collaborative Agreement Number W911NF-18-2-0048. This work is also supported in part by the the National Science Foundation under Cooperative Agreement PHY-2019786 (The NSF AI Institute for Artificial Intelligence and Fundamental Interactions, http://iaifi.org/).

## REPRODUCIBILITY STATEMENT

Algorithm 1, the original public code for each of the I-SSL methods we use in the paper and the experimental setups in Section 4.1, and in Appendices D, E and F, can be used for reproducibility. Our code is available at https://github.com/rdangovs/essl.

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

## A SUMMARY OF MAIN TEXT AND LAYOUT OF APPENDIX

In this paper we motivated the generalization of state-of-the-art methods in self-supervised learning to the more general framework of equivariant self-supervised learning (E-SSL). In E-SSL rather than using only invariance as a trivial case of equivariance, we encouraged non-trivial equivariance and improved state-of-the-art methods on common computer vision benchmarks and regressions tasks in photonics science. We also discussed that there are many types of equivariance we can consider for E-SSL. We observed that most of the successful transformations for E-SSL that we explored form groups, but forsee that potentially many more transformations could be explored.

For future work one could learn transformations that are equivariances, instead of setting them manually. Thus, the concept of E-SSL could potentially be extended to natural language processing or other science domains, whose transformations for SSL are less well-understood. To facilitate further research in E-SSL, below we provide additional details and analysis of the experiments in the main text. We also discuss interesting avenues for future work.

## B PROOF OF PROPOSITION 1

*Proof.* To construct a non-trivially equivariant $f$, we first need to show that both $X'$ and $S$ are $G$-sets, i.e. that there is a group action $T_g$ of $G$ on $X'$, which is given by the statement of the proposition, and another (non-trivial) group action $T'_g$ of $G$ on $S$, which we will construct. Then, we need to show that $f$ commutes with the group action, i.e. that $f(T_g(\boldsymbol{x}')) = T'_g(f(\boldsymbol{x}'))$.

**Group actions.**   Note that by the setup of the problem, we are already given how $G$ acts on the input $X'$, i.e. $T_g$ is known. For example, if $G$ is the group of four-fold rotations, then $T_g$ is the rotation of the input by a multiple of $\pi/2$. We proceed to construct the non-trivial group action $T'_g$ of $G$ on $S$.

Define the function $T' \colon G \times S \to S$ as $T'(g, s) = f(T_g(T_{g'}(\boldsymbol{x}')))$, where $s = f(T_{g'}(\boldsymbol{x}'))$. Note that $T'$ is well-defined, because $gg' \in G$ by the closure of the group and $s$ is uniquely written as $s = f(T_{g'}(\boldsymbol{x}'))$. To see why $s$ is uniquely written, it suffices to show that if $f(T_{g'}(\boldsymbol{x}')) = f(T_{g''}(\boldsymbol{x}''))$ then both $g' = g''$ and $\boldsymbol{x}' = \boldsymbol{x}''$, which follows directly from our assumption in the statement.

Now, to prove that $T'$ is a group action, it suffices to show two properties.

- Identity: $T'(e, s) = s$ for $s = f(g'(\boldsymbol{x}'))$ and $e$ is the unit element of the group. To show that, note that by definition $T'(e, s) = f(T_e(T_{g'}(\boldsymbol{x}'))) = f(T_{g'}(\boldsymbol{x}'))$, because $eg' = g'$.

- Compositionality: $T'(g, T'(h, f(T_{g'}(\boldsymbol{x}')))) = T'(gh, f(T_{g'}(\boldsymbol{x}')))$. To show this, we expand the LHS and use the definition of $T'$ to obtain as follows $T'(g, T'(h, f(T_{g'}(\boldsymbol{x}')))) = T'(g, f(T_h(T_{g'}(\boldsymbol{x}')))) = f(T_g(T_{hg'}(\boldsymbol{x}'))) = f(T_{ghg'}(\boldsymbol{x}')) = f(T_{gh}(T_{g'}(\boldsymbol{x}'))) = T'(gh, f(T_{g'}(\boldsymbol{x}')))$, because the group operation is associative.

Hence, $T'$ is a group action, and thus $S$ is a $G$-set, and we can write $T'(g, \cdot) \equiv T'_g(\cdot)$.

**Commuting with the group action.**   To see this property, note that $T'_{g'}(f(\boldsymbol{x}')) = T'_{g'}(f(T_g(\boldsymbol{x}))) = f(T_{g'}(T_g(\boldsymbol{x}))) = f(T_{g'}(\boldsymbol{x}'))$ as desired. Note that $T'_g$ is non-trivial.

Therefore, we can conclude that $f$, which satisfies the constructed group action $T'_g$, is not-trivially equivariant to the group $G$. ☐

## C   ROTATION PREDICTION AND I-SSL BENEFIT FROM SIMILAR DATA AUGMENTATION.

Recently, rotation prediction with a linear head from the frozen backbone representations proved to be useful for validating the augmentation policies of contrastive learning (Reed et al., 2021). This shows that the two tasks of classification of *ground truth classes* and *synthetic rotation classes* from frozen backbone representations benefit from similar augmentation policies. We took this experiment a step further, and performed rotation prediction with the augmentation policies, typically used in contrastive learning.

The result is in Table 5. Interestingly, RotNet benefits from augmentations, typically used in contrastive learning, and the RotNet training shares the same sweet spot (Tian et al., 2020) as kNN classification. There are several takeaways from this experiment: (*i*) we can find good augmentations for contrastive learning by doing RotNet *alone*, i.e. without doing *any* contrastive learning; (*ii*) RotNet benefits from augmentations needed in contrastive learning; (*iii*) we may be able to combine four-fold rotations prediction and contrastive learning.

## D   CIFAR-10 EXPERIMENTS

### D.1   EXPERIMENTAL SETUP

Our experiments use the following architectural choices: ResNet-18 backbone (the CIFAR-10 version has kernel size 3, stride 1, padding 1 and there is no max pooling afterwards); 512 batch size (only our baseline SimSiam model uses batch size 1024); 0.03 base learning rate for the baseline SimCLR and SimSiam and 0.06 base learning rate for E-SimCLR and E-SimSiam; 800 pre-training epochs; standard cosine decayed learning rate; 10 epochs for the linear warmup; two layer projector with hidden dimension 2048 and output dimension 2048; for SimSiam a two layer (bottleneck) predictor with hidden dimension 512 whose learning rate is not decayed; the last batch normalization for the projector does not have learnable affine parameters; 0.0005 weight decay value; SGD with momentum 0.9 optimizer. The augmentation is Random Resized Cropping with scale (0.2, 1.0),

Table 5: RotNet's augmentation sweet spot. kNN and Rotation Prediction have the same sweep spot (Level 4) which gives best accuracy in both columns. RotNet is trained on CIFAR-10 for 100 epochs with the same optimization setup as in our I-SSL experiments. Accuracies are on the test split. ($\downarrow \cdot$) marks the deviation from the sweet spot. Every new level adds a new augmentation to the previous level incrementally.

| Level | Added Augmentation | Supervised kNN Acc. (%) | Rotation Prediction Acc. (%) |
|---|---|---|---|
| 0 | none | 44.8 ($\downarrow$ 19.8) | 90.2 ($\downarrow$ 4.8) |
| 1 | random resized cropping | 59.2 ($\downarrow$ 5.4) | 93.7 ($\downarrow$ 1.3) |
| 2 | horizontal flips w.p. 0.5 | 59.4 ($\downarrow$ 5.2) | 94.5 ($\downarrow$ 0.5) |
| 3 | color jitter w.p. 0.8 | 64.3 ($\downarrow$ 0.3) | 94.9 ($\downarrow$ 0.1) |
| 4 | grayscale w.p. 0.2 | **64.6** | **95.0** |
| 5 | Gaussian blur w.p. 0.2 | 64.1 ($\downarrow$ 0.5) | 94.5 ($\downarrow$ 0.5) |
| 6 | random rotation ($\pm\pi/6$) | 59.4 ($\downarrow$ 5.2) | 93.1 ($\downarrow$ 1.9) |
| 7 | vertical flip w.p. 0.5 | 51.9 ($\downarrow$ 12.7) | 90.6 ($\downarrow$ 4.4) |

aspect ratio (3/4, 4/3) and size 32x32, Random horizontal Flips with probability 0.5, Color Jittering (0.4, 0.4, 0.4, 0.1) with probability 0.8 and Grayscale with probability 0.2. Some of our evaluations use a kNN-classifer with 200 neighbors, cosine similarity and Gaussian kernel with temperature 0.1. This evaluation correlates well with the standard linear probe, but it is more efficient to calculate. We report the kNN accuracy in % at the end of the 800 epochs of training. For our main results, we report a linear probe accuracy from training a linear classifier for 100 epochs on top of the frozen representations with SGD with momentum 0.9 and cosine decay of the learning rate, batch size 256 and initial laerning rate of 30. For linear probe experiments we try 5 different initalizations of the linear head and report mean and standard deviations. The deviations are negligible because the linear probe is robust to the random seed. All parameters are reported in a Pytorch-like style.

For Figure 1 we use resolution of 32x32 for the transformations studied. The 4 levels of the Gaussian blur are for kernel sizes 0, 5, 9 and 15 in the default Gaussian blur torchvision implementation. The prediction of the transformations follows the experimental setup in Section 3. When we apply the transformations in I-SSL, we add them in the beginning of the augmentation policy with probability 1. The same setup is used for "Disentangled representations" and "Insensitive instead of sensitive" in Table 1.

## D.2 ADDITIONAL EXPERIMENTS

**Explored hyperparameters.** Both for SimCLR and SimSiam we ran a grid search over the following hyperparameters: base learning rate: {0.01, 0.03, 0.06}, batch size: {512, 1024}, $\lambda$ (for E-SSL): {0.0, 0.2, 0.4, 0.6, 0.8, 1.0}, predictor's MLP depth: {2, 3, 4}, predictor's normalization: {None, BatchNorm, LayerNorm}, nonlinearity at the last MLP layer of the predictor: {True, False}.

**Tuning $\lambda$.** Table 6 shows tuning of the CIFAR-10 results. We observe noticeable improvements over the SSL baselines by using E-SSL instead.

Table 6: Tuning the $\lambda$ parameter for CIFAR-10.

| Method | Baseline | E-SSL | | | | |
|---|---|---|---|---|---|---|
| | 0.0 | 0.2 | 0.4 | 0.6 | 0.8 | 1.0 |
| SimCLR | 92.0$_{\pm 0.0}$ | 93.6$_{\pm 0.0}$ | **94.1**$_{\pm 0.0}$ | 94.0$_{\pm 0.0}$ | **94.1**$_{\pm 0.0}$ | 93.5$_{\pm 0.0}$ |
| SimSiam | 91.1$_{\pm 0.0}$ | 94.1$_{\pm 0.0}$ | **94.2**$_{\pm 0.1}$ | 93.7$_{\pm 0.0}$ | 93.8$_{\pm 0.0}$ | 93.3$_{\pm 0.0}$ |

**Sensitivity to transformations for I-SSL.** Table 7 demonstrates that E-SSL can produce good representation with as few SSL transformations for I-SSL as possible. We observe that E-SSL is less sensitive than SSL to the choice of data augmentation.

Table 7: Comparing the augmentation sensitivity for CIFAR-10. Levels: 0 is no transformations; 1 adds random resized cropping; 2 adds horizontal flips; 3 adds color jitter; 4 adds grayscale.

| Method | Augmentation Level | | | | |
|---|---|---|---|---|---|
| | 0 | 1 | 2 | 3 | 4 |
| SimCLR | $28.0_{\pm 0.1}$ | $76.0_{\pm 0.1}$ | $77.0_{\pm 0.0}$ | $87.4_{\pm 0.0}$ | $92.0_{\pm 0.0}$ |
| E-SimCLR | $69.5_{\pm 0.0}$ | $83.5_{\pm 0.0}$ | $84.6_{\pm 0.1}$ | $91.6_{\pm 0.0}$ | $\mathbf{94.1}_{\pm 0.0}$ |
| SimSiam | $17.6_{\pm 0.1}$ | $71.5_{\pm 0.0}$ | $72.2_{\pm 0.0}$ | $88.1_{\pm 0.0}$ | $91.1_{\pm 0.0}$ |
| E-SimSiam | $67.5_{\pm 0.1}$ | $84.6_{\pm 0.0}$ | $85.7_{\pm 0.1}$ | $92.9_{\pm 0.0}$ | $\mathbf{94.2}_{\pm 0.1}$ |

**The importance of complete invariance or sensitivity.** Table 8 studies whether a middle ground for the representations exist, i.e. whether it is possible to have part of the representation invariant and the other part sensitive to the transformation. If we apply the E-SSL loss only to half of the representation, then there is a very small drop in the performance. Furthermore, we observe that having a disjoint mix between insensitivity and sensitivity in the representation is noticeably harmful.

Table 8: Studying the effect of disjoint representations on CIFAR-10. Split Representation means that we encourage similarity only on one half of the backbone representation. Disentangled Representation means that one half of the representation is trained to be insensitive to four-fold rotations and the other half is sensitive four-fold rotations. Linear probe accuracy (%) after 800 epochs.

| Method | Baseline | Split Representation | Disentangled Representation |
|---|---|---|---|
| E-SimCLR | $\mathbf{94.1}_{\pm 0.0}$ | $94.1_{\pm 0.0}$ ($\downarrow 0.0$) | $91.3_{\pm 0.0}$ ($\downarrow 2.7$) |
| E-SimSiam | $\mathbf{94.2}_{\pm 0.1}$ | $93.8_{\pm 0.0}$ ($\downarrow 0.4$) | $91.1_{\pm 0.0}$ ($\downarrow 3.1$) |

**Fully connected backbone.** We perform a simple experiment with a fully connected backbone, instead of a ResNet-18. The hidden dimensions of the backbone are listed in order as $\{3\times32\times32,$ 2048, 2048, 512\}$ with Batch Normalization and ReLUs in between. The rest of the experimental setup is exactly the same. On the linear probe (%), we obtain $70.5\pm0.0$ for SimCLR and $73.8\pm0.1$ for E-SimCLR, and $70.9\pm0.0$ for SimSiam and $73.5\pm0.1$ for E-SimSiam, highlighting noticeable gains from using E-SSL.

**CIFAR-100 experiments.** We test our CIFAR-10 experimental setup directly on CIFAR-100. On the linear probe (%), we obtain $65.8\pm0.0$ for SimCLR and $69.5\pm0.1$ for E-SimCLR, and $65.8\pm0.1$ for SimSiam and $69.3\pm0.1$ for E-SimSiam, highlighting sizable gains from using E-SSL.

**Large crop study.** We study whether using a large crop with a single rotation on CIFAR-10 can be just as good as a small crop. We obtain $93.9\pm0.0$ on the linear probe using E-SimCLR, which is only 0.2 absolute points below our best result of $94.1\pm0.0$ using four small crops.

### D.3 NORM-DIFFERENCES ANALYSIS

In Figure 6 we present analysis that shows our training objectives encourage invariance and equivariance to transformations. We take our best performing E-SimCLR and E-SimSiam methods on CIFAR-10. During training we keep track of two measures that can capture how invariant/ equivariant the backbone representations are.

The "invariance measure" computes the negative cosine similarity between two views of the backbone representations. The lower this measure is, the higher the similarity between the two views, and thus the more invariant the backbone representations are to the transformations in I-SSL. We observe that during training high similarity between the two views is maintained (roughly between 0.8 and 0.9), which indicates that invariance is encouraged in the backbone representations, as desired.

Likewise, the "equivariance measure" computes the average cosine similarity of the backbone representations, between all six pairs of the four rotated views. The lower this measure is, the lower

the similarity between the four views, and thus the more non-trivially equivariant the backbone representations are to the transformations for equivariance. We observe that the measure decays to about 0.3 during training, which indicates that the backbone representations are encouraged to be equivariant to four-fold rotations, as desired.

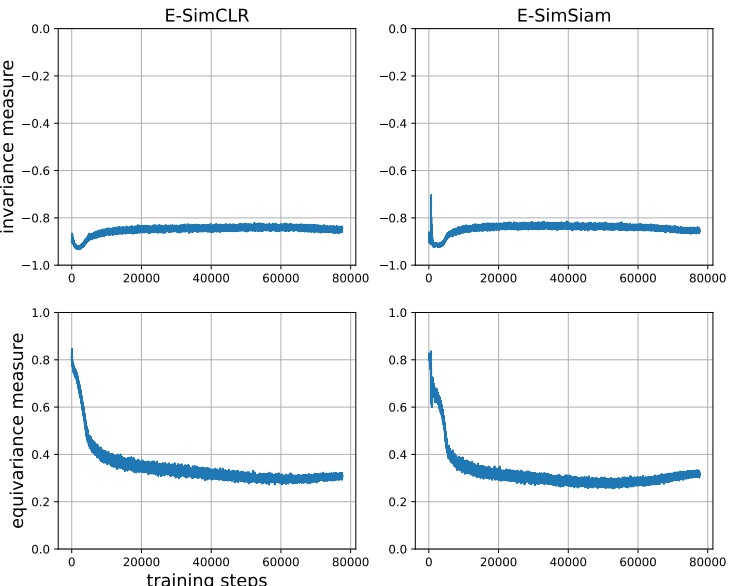

Figure 6: Demonstration of the evolution of the invariance (top) and equivariance (bottom) measures during training. Left is E-SimCLR and right is E-SimSiam.

# E  IMAGENET EXPERIMENTS

We had limited computational resources, so we kept the learning rates the same as in the original methods. Only for SimSiam we found that choosing a smaller learning rate 0.08 leads to better results for E-SimSiam. We only swept the $\lambda$ parameter, where for SimCLR and SimSiam the sweep was between 0 and 1 and for Barlow Twins it was between 0 and 100. The optimal $\lambda$ is 0.4 for SimCLR, 0.08 for SimSiam, 8 for Barlow Twins. We use (0.05, 0.14) scale range for 100 pre-training epochs. For more pre-training epochs we use (0.05, 0.14) for SimCLR and (0.08, 1.0) for Barlow Twins.

Table 9 lists the overhead from using rotation prediction in our experiments.

Table 9: Overhead in doing rotation prediction. Reported GPU hours for an experiment on 100 epochs.

|  | SimCLR | SimSiam | Barlow Twins |
| --- | --- | --- | --- |
| Baseline | 256 | 295 | 246 |
| E-SSL (ours) | 307 | 364 | 294 |
| Overhead | 20% | 23% | 19% |

## F  PHC EXPERIMENTS

**Dataset generation.**    2D Photonic crystals (PhCs) are characterized by a periodically varying permitivitty $\varepsilon(x, y)$; here, for simplicity we consider a "two-tone" permitivitty profile i.e. $\varepsilon \in \{\varepsilon_1, \varepsilon_2\}$, with $\varepsilon_i \in [1, 20]$ discretized to a resolution of $32 \times 32$. To generate the unit cells in the "blob" dataset, we follow the proceedure in Christensen et al. (2020). For the Gpm dataset, the unit cells are defined using a level set of a 2D Fourier sum function like in Kim et al. (2021); Loh et al. (2021), with additional constraints applied to the lattice to create the mirror symmetry adopted from the method in Christensen et al. (2021). We then follow the procedure in Loh et al. (2021) to compute, and subsequently process, the density-of-states (DOS) of each unit cell, specifically, via the MIT Photonics Bands (MPB) software (Johnson & Joannopoulos, 2001) and the Generalized Gilat-Raubenheimer method in an implementation from Liu et al. (2018).

**Network architecture.**    We use an encoder network composing of simple convolutional (CNN) and fully-connected (FC) layers for the backbone; specifically, our backbone begins with 3 CNN layers, all with a kernel size of 7 and channel dimensions given by $[64, 256, 256]$. The output is flattened and fed into 2 FC layers each with 1024 nodes (i.e. the representations have dimension 1024). We include BatchNorm (Ioffe & Szegedy, 2015), ReLU and MaxPooling for the CNNs, and ReLU only for the first FC layer. The projector and predictor networks, $p_1$ and $p_2$ are 2-layer MLPs with hidden dimension 512, with BatchNorm and ReLU between each layer except the last and the projection dimension for $p_1$ is 256. Additionally, since this is a regression task and the label space is much larger than in image classification tasks, we include a dense DOS-predictor head after the representations, which is fine-tuned with 3000 labelled samples after SSL or E-SSL. The DOS-predictor has 4 FC layers, with number of nodes given by $[1024, 1024, 512, 400]$. We explore two fine-tuning protocols of the DOS-predictor: freezing the backbone (discussed later in the Appendix) or fine-tuning the backbone (discussed in the main text).

**Hyperparameters.**    For SSL and E-SSL, we performed 250 pre-training epochs using the SGD optimizer with a standard cosine decayed learning rate; the batch size was fixed to 512. The pre-trained model was saved at various epochs $\{20, 50, 100, 180, 250\}$ for further fine-tuning. Fine-tuning was performed for 100 epochs using Adam optimizer and a fixed batch size of 64. No transformations were applied to the input during fine-tuning for both freezing or fine-tuning the backbone. We ran a grid search over the following hyperparameters; for pre-training, base learning rate: $\{10^{-3}, 10^{-4}, 10^{-5}\}$, $\lambda$ (for E-SSL): $\{0.2, 1.0, 2.0, 5.0, 10.0\}$, and for fine-tuning: a learning rate in $\{10^{-3}, 10^{-4}, 10^{-5}\}$.

**Frozen backbone experiment.**    In Table 10 we present our results from freezing the backbone encoder while fine-tuning the DOS-predictor head. We observe similar trends as in Table 4 where we allowed fine-tuning of the backbone. Relative error is reported in % and the lower the error is, the better. SimCLR for Blob includes $C_{4v}$ (rotations and flips) and SimCLR for Gpm includes rolling translations and flips. E-SimCLR encourages the features to be sensitive to the selected transformation (four-fold translations for Blob and four-fold rotations for Gpm), which improves the performance of SimCLR. On the contrary, adding the selected transformation to SimCLR, as indicated by "+ Transform", increases the error of SimCLR. Error bars are reported for 3 different choices of training data. Supervised (frozen) refers to the impractical situation of freezing a random backbone and fine-tuning the DOS-predictor.

Table 10: Frozen backbone experiment on PhC datasets for 3000/ 2000 labelled train/ test samples.

| PhC Dataset | Supervised (frozen) | SimCLR | SimCLR + Transform | E-SimCLR (ours) |
|---|---|---|---|---|
| Blob | $1.686_{\pm\ 0.014}$ | $1.237_{\pm\ 0.005}$ | $1.242_{\pm\ 0.013}$ | $\mathbf{1.165}_{\pm\ 0.020}$ |
| Gpm | $5.450_{\pm\ 0.077}$ | $3.214_{\pm\ 0.048}$ | $3.313_{\pm\ 0.029}$ | $\mathbf{3.187}_{\pm\ 0.000}$ |

**Continuous group experiment.**    In all experiments shown so far, we dealt with finite groups of transformations. To show that E-SSL generalizes beyond the finite group setting, we also explore transformations from a continuous group. An example is the scaling transformation where every pixel of the input unit cell is scaled by the same positive factor. More specifically, this set of positive

scaling transformations $g(s)\boldsymbol{x} = s\boldsymbol{x}$ defines a continuous group $G = \{g(s)|s \in \mathbb{R}_+\}$ which leaves the DOS labels invariant due to the physics of the problem and normalization applied when pre-processing the dataset (Loh et al., 2021). In our experiment, we uniformly sample $s \in (1, s_{max}]$ and apply the inverse with probability 0.5 (i.e. we cap the scaling factor to a maximum of $s_{max} = \{5, 10\}$ for numerical stability during training and we apply up-scaling and down-scaling with equal probability). To encourage equivariance to this group, we simply predict the scale factor applied to the input using L1 loss (i.e. the final layer of the predictor $p_2$ is a single node). In Table 11, we show results after fine-tuning the backbone and the DOS predictor network with 3000 labelled samples. We observe similar trends to Table 4; encouraging sensitivity to scaling produces the lowest error and including scaling to SimCLR increases the error. To isolate the effect of scaling transformation, the remaining physics-governed invariances excluding scaling (translations, rotations and mirrors) are used in SimCLR and the invariance part of E-SimCLR for both datasets.

Table 11: Fine-tuning the backbone on PhC datasets using 3000/ 2000 labelled train/ test samples. Relative error (%) is $\ell_{\text{DOS}} = (\sum_\omega |\text{DOS}^{\text{pred}} - \text{DOS}|)/(\sum_\omega \text{DOS})$. Lower is better. E-SimCLR encourages the features to be sensitive to scaling. "+ Scaling" means adding scaling to SimCLR. Error bars are for 3 different training data splits.

| PhC Dataset | Supervised | SimCLR | SimCLR + Scaling | E-SimCLR (ours) |
|---|---|---|---|---|
| Blob ($s_{max} = 10$) | $1.068_{\pm 0.015}$ | $0.988_{\pm 0.001}$ | $1.005_{\pm 0.006}$ | $\mathbf{0.974}_{\pm 0.000}$ |
| Blob ($s_{max} = 5$) | $1.068_{\pm 0.015}$ | $0.988_{\pm 0.001}$ | $1.000_{\pm 0.014}$ | $\mathbf{0.987}_{\pm 0.017}$ |
| Gpm ($s_{max} = 10$) | $3.212_{\pm 0.041}$ | $3.073_{\pm 0.003}$ | $3.112_{\pm 0.011}$ | $\mathbf{3.062}_{\pm 0.005}$ |
| Gpm ($s_{max} = 5$) | $3.212_{\pm 0.041}$ | $3.073_{\pm 0.003}$ | $3.082_{\pm 0.013}$ | $\mathbf{3.058}_{\pm 0.008}$ |

## G    FLOWERS-102 EXPERIMENTS

In order to study the importance of the assumption in Proposition 1 we perform an experiment with a dataset that might not be amenable to such an assumption at first sight. We choose the Flowers-102 dataset (Nilsback & Zisserman, 2008), because at first sight the dataset might not benefit from four-fold rotations in E-SSL. Therefore, this dataset complements the experiments we performed for CIFAR-10 and ImageNet.

**Experimental setup.**    We train SimCLR and E-SimCLR. We use the same optimization hyperparameters from the experimental setup for our CIFAR-10 experiments. We downsize the images to 96x96 resolution and use the standard ResNet-18, instead of its modified version for CIFAR-10. For the data augmentation in I-SSL, we use the same RandomResizedCropping as in CIFAR-10 (with size 96 of the crops being the only difference), the same Color Jittering and Random horizontal flips as in the CIFAR-10 experiment. We report the kNN accuracy in (%) on the validation set. We study both four-fold rotations and four-fold translations as transformations for invariance/ equivariance. Four-fold rotations are chosen following the hypothesis that most of the data points should be invariant to rotation. Four-fold translations are chosen, because of our observation that most of the data points are centered, just like in the Blob PhC dataset. The $\lambda$ for predicting four-fold translations is 0.01 and for four-fold rotations is 0.5 (chosen from a grid search among $\{0.001, 0.01, 0.1, 0.5, 1.0, 2.0\}$).

**Results.**    Following our observations in Figure 1, we observe that encouraging insensitivity to four-fold rotations and translations, by adding the transformations to the SimCLR data augmentation, worsens the SimCLR baseline. In contrast, using these transformations for E-SSL improves the baselines and further shows the utility of E-SSL for real-world data. We even observe benefit from encouraging equivariance to four-fold rotations, which is against the intuition that rotations should be invariant. This is probably due to the fact that some images in the dataset are not truly rotationally invariant (see examples of the data points in Figure 8).

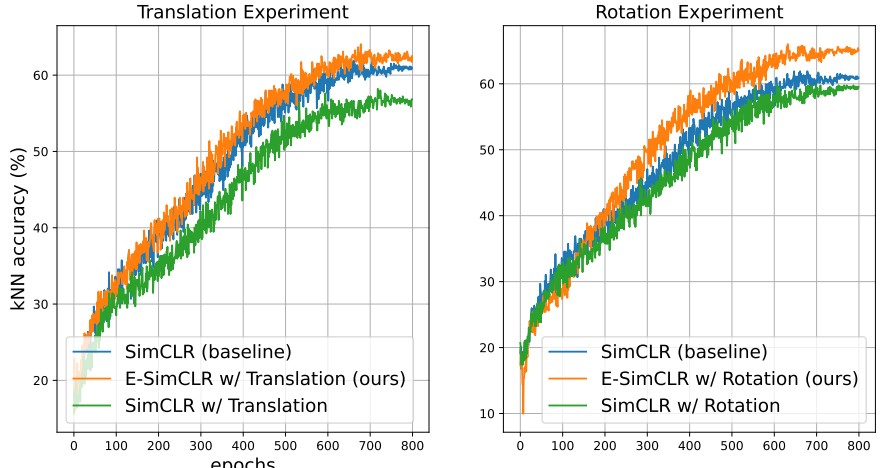

Figure 7: E-SimCLR gives sizable improvements for the Flowers-102 SSL pre-training. kNN accuracy (%) is on the validation set.

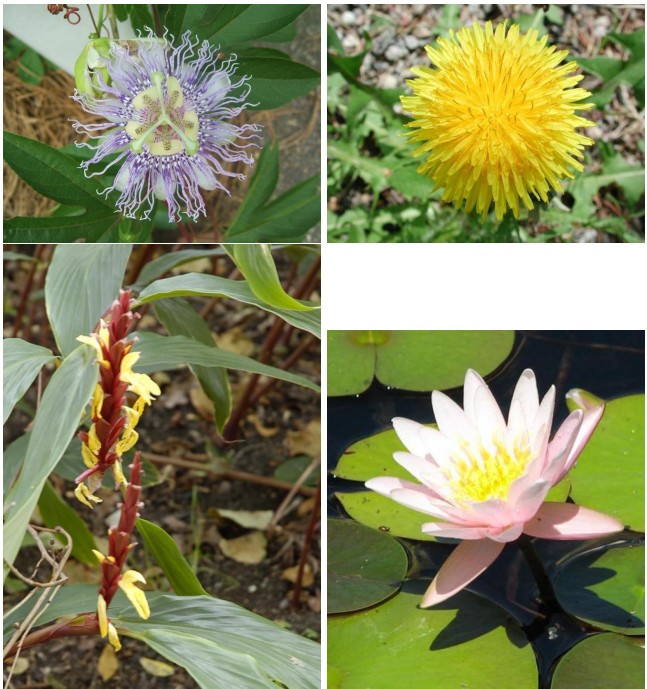

Figure 8: The Flowers-102 is not completely invariant to rotation. The top row shows data points which are roughly invariant to four-fold translations. The bottom row shows counterexamples to that hypothesis.

## H  RELATIVE ORIENTATION PREDICTION

In our experiments we demonstrate that if a shared biased between the train and test sets exists, we should exploit it via the E-SSL training objective. However, in some scenarios, the class label of the downstream tasks depends on the orientation (e.g., classifying road signs) and the current E-SSL method may not be very useful, because both $x$ and $T_g(x)$ exist in the data. This situation

invites a natural generalization of our method in the spirit of (Agrawal et al., 2015). If $x$ and $T_g(x)$ exist in the data then we can modify E-SSL minimally to form a useful objective. In particular, we can set the objective for $p_2$ to predict the relative orientation between two data points, i.e. given $x$, we form $T_{g'}(x)$ by sampling $T_{g'}$ from $G$ uniformly, and then predict $g'$ from $p_2(z; \theta_{p_2})$, where $z = [f(x), f(T_{g'}(x))]$ is the concatenation of the two representations. This modification requires minimal change to our framework.

To test the usefulness of the modified method when $x$ and $T_{g'}(x)$ exist in the data, we artificially modify CIFAR-10 so that any rotation of an image can appear in the dataset. We consider the downstream task of predicting the rotation orientation of an image, which clearly depends on the orientation of the image. Our hypothesis is that the modified E-SimCLR will be better than SimCLR, which is what we observe in our results below. Intuitively, SimCLR is not able to capture the orientation of the image, while the modified E-SimCLR is able to, because the latter predicts relative orientation.

The experimental setting is as follows: we pretrain for 100 epochs. The predictor for equivariance's input dimension is doubled, because we concatenate two representations. We set $\lambda$ to 0.4. All other hyperparameters are the same as the rest of the CIFAR-10 experiments. The downstream task is 4-way rotation orientation classification. Using pre-training with SimCLR on the standard CIFAR-10 dataset, we obtain baseline linear probe (%) accuracy 67.1±0.1. Using our modification of E-SSL on the same experimental setting, we obtain **71.2±0.1**. linear probe accuracy, which is a sizable gain and points to the promise of the relative orientation prediction as future work.

The relative orientation prediction scenario is also highly relevant in other domains such as in photonic crystals. For example, we can modify the PhC setup and remove the "orientation bias" of the datasets while predicting a different property, the band structures. Unlike the DOS, the PhC band structures are not invariant to rotations and so we can consider the group of four-fold rotations. This setup would fit the scenario described above since 1) both $x$ and $T_{g'}(x)$ exist in the data due to the lack of bias and 2) the downstream task is sensitive to rotation. We will explore the above framework on this setup in future work.

