# OpenReview forum: "Equivariant Self-Supervised Learning: Encouraging Equivariance in Representations"
_ICLR.cc/2022/Conference — ICLR 2022 Poster_

### Official Review · Reviewer_p4FW · 2021-11-01

**Correctness:** 3
**Technical Novelty And Significance:** 2
**Empirical Novelty And Significance:** 3
**Recommendation:** 6
**Confidence:** 4

**Main Review:**

# A. Strengths
- 1.	The proposed approach achieves good performance gain over baselines on CIFAR10 and ImageNet.
- 2.	The proposed approach generalizes the existing invariant SSL approach to also include equivariance; the idea is intuitive to follow and reasonable.
- 3.	Author promised to release code and the supplemental materials reports the hyperparameters sweeps and results.

# B. Weaknesses
- 1.	Novelty & Contribution: The approach can be viewed as combining “predicting image rotation” (Gidaris et al. 2018) with existing invariant SSL. The authors should more clearly explain the differences in the related works section. Also, the work proposes a “general framework”, I find the writing which only focuses on four-fold rotation not adequate. I would recommend writing out the generalized formulation in more detail, e.g. using equivariance, and not just a very abstract Eq. 1.
- 2.	Limitation in approach: The proposed approach is limited to finite groups as it is based on classification. Has the author thought about further generalization?
- 3.	Organization: The analysis in Sec. 5.2 seems out of place. Why not include it in the approach section? Also, personally, I feel the related works before the discussion does not help with the organization. I would recommend placing it after the introduction.
- 4.	Clarity: The link between equivariance and the proposed approach is not clearly explained in the paper. As the authors have introduced the equivariance definition and notation, it might be beneficial to use those in their approach. Next, the drawing of Figure. 3 should be improved. For example, I cannot understand why for the equivariance there is only one backbone f; for the invariance there are two?. Also, having a single arrow going into “equivariance” is very confusing.
- 5.	Ablations in Table 1: Can the author explain “Linear enhancing predictor” and “No SSL augmentation in the enhanced views” in the ablation table. Those lack a clear description in the text and it's not included in Algorithm 1.
- 6.	Additional ablation studies.
  - a)	The authors should investigate whether the learned representation is indeed equivariant/invariant. For example, computing norm-differences between the features. This would verify, empirically, whether the desired features are learned.
  - b)	I would like to see an ablation on the “crop with size and scale” for the different prediction views. I understand that it is used to save memory; however, I wonder if it has an impact on performance. For example, in Fig. 3 the crop nicely centers on the object and removes background. Please conduct an experiment without the crop/resize, to make it feasible for the memory, just uniformly sample the rotation during training, i.e., an unbiased estimate of Eq. 2.

# C. Misc.
- Maybe move Alg. 1 and Table 3 to the top of the page; I think this would make the paper neater.
- Might consider citing the following related works:
  - Misra, Ishan, C. Lawrence Zitnick, and Martial Hebert. "Shuffle and learn: unsupervised learning using temporal order verification." Proc. ECCV. 2016
  - Lee, Hsin-Ying, et al. "Unsupervised representation learning by sorting sequences." Proc. ICCV. 2017
  - Mundhenk, T. Nathan, Daniel Ho, and Barry Y. Chen. "Improvements to context based self-supervised learning." Proc. CVPR, 2018.


**Summary Of The Paper:**

This paper proposes a framework which generalizes self-supervised learning (SSL) to also learn equivariance behavior. A family of SSL (the authors named invariant SSL) encourages representations which learn features that are invariant to certain transformations, e.g., horizontal flips. As invariance is a special case of equivariance, this paper explores whether encouraging equivariant representation is beneficial. Specifically, they focused on four-fold rotations and showed that combined with existing SSL methods lead to improvement in classification performance on CIFAR10 and ImageNet. The author also showcases application to photonics science.

**Summary Of The Review:**

My main concern is with the writing and novelty. The paper proposes a general framework. However, it is written as a combination of invariant SSL + four-fold rotation; see additional suggestions in weaknesses section. The authors should better highlight their novelty and differences from existing work. Hence, I recommend a weak reject.

---

> ### Author Response · Authors · 2021-11-17
> **1. An important new experiment on an infinite group; 2. Clarifying novelty; 3. Improving organization and clarity.**
>
> > Novelty & Contribution: The approach can be viewed as combining “predicting image rotation” (Gidaris et al. 2018) with existing invariant SSL. The authors should more clearly explain the differences in the related works section. Also, the work proposes a “general framework”, I find the writing which only focuses on four-fold rotation not adequate. I would recommend writing out the generalized formulation in more detail, e.g. using equivariance, and not just a very abstract Eq. 1.
>
> Thank you for your suggestions that helped us underscore our novelty! We reorganized that part of the paper that introduces the method and wrote out the general form of Eq. 1 as a prediction loss that encourages equivariance. Please note that our work studies many more transformations beyond rotation prediction, as we discuss throughout our rebuttal. We chose to use rotation prediction for equivariance on CIFAR-10 and ImageNet experiments, because readers are familiar with this transformation. Thus, our choice serves as an intuitive introduction to the E-SSL method. In order to underscore our novelty, we updated our introduction (contributions bullet points) and in the related work we clearly compared our method against related works that encourage equivariance to transformations.
>
> > Limitation in approach: The proposed approach is limited to finite groups as it is based on classification. Has the author thought about further generalization?
>
> Thanks for this suggestion. We have now updated our paper to include an experiment to show that E-SSL can also be generalized to infinite groups, which is shown in the “continuous group experiment” section in Appendix E. More specifically, we demonstrate this on the PhC DOS prediction problem discussed in Section 5. Due to the physics of the problem and the normalization of the labels, the DOS labels are invariant when an overall positive scale factor is applied to the input and this set of positive scale factors defines a continuous group. For encouraging equivariance, instead of cross-entropy loss used in finite groups, we use L1 loss to predict the scale transformation applied and we show in Table 11 of Appendix E that E-SSL is indeed also effective for continuous groups.
>
> > Organization: The analysis in Sec. 5.2 seems out of place. Why not include it in the approach section? Also, personally, I feel the related works before the discussion does not help with the organization. I would recommend placing it after the introduction.
>
> We appreciate your suggestions for helping us improve the organization! We updated the paper accordingly.
>
> > Clarity: The link between equivariance and the proposed approach is not clearly explained in the paper. As the authors have introduced the equivariance definition and notation, it might be beneficial to use those in their approach. Next, the drawing of Figure. 3 should be improved. For example, I cannot understand why for the equivariance there is only one backbone f; for the invariance there are two?. Also, having a single arrow going into “equivariance” is very confusing.
>
> Thank you for your suggestions! There is a single backbone f and arrow for equivariance, because the prediction views are grouped in the same batch (for efficiency reasons). However, it is much clearer to present it with four backbone fs and four arrows. Please, refer to the updated figure and let us know if you would like us to make further changes. Thanks!
>
> > Ablations in Table 1: Can the author explain “Linear enhancing predictor” and “No SSL augmentation in the enhanced views” in the ablation table. Those lack a clear description in the text and it's not included in Algorithm 1.
>
> Thank you for catching this. We mean “Linear predictor for equivariance” and “No SSL augmentation in equivariance views”. The former experiment shows that a linear predictor is suboptimal than a non-linear one, and the latter experiment shows that typical SSL augmentation (i.e. the standard SimCLR augmentation strategy) is useful for the prediction objective. The latter point is also consistent with the experiment in Appendix B. To our knowledge, we are the first in the literature to observe the importance of using augmentation in contrastive learning for the rotation prediction task. We updated the paper accordingly.

---

> > ### Comment · Reviewer_p4FW · 2021-11-23
> > **Reviewer Response**
> >
> > Thanks for the response and updating the paper. The additional results and rewriting have clarified the paper's contribution.
> > I have increased my recommendation.

---

> ### Author Response · Authors · 2021-11-17
> **4. New ablations on (i) norm-difference-measure to capture invariance/ equivariance and (ii) crop with size and scale; 5. Misc stuff.**
>
> > a) The authors should investigate whether the learned representation is indeed equivariant/invariant. For example, computing norm-differences between the features. This would verify, empirically, whether the desired features are learned.
>
> Yes, we agree that this is a useful study and we addressed that in Appendix C.3. We can easily verify the equivariant/invariant features encouragement. We can use SimSiam’s loss on the backbone representations of the two I-SSL views to verify that the invariance is encouraged. SimSiam’s loss is the negative cosine similarity between the two views. This means that to encourage invariance we would like to see a larger cosine similarity between the views. We see that both E-SimCLR and E-SimSiam encourage a large cosine similarity (about 0.8).
>
> In a similar fashion, we can use cosine similarity between differently transformed input views for equivariance to measure whether equivariance is encouraged. The objective for equivariance is that the cosine similarity between pairs of transformed views should be *smaller* (notice the difference with invariance). We see that both E-SimCLR and E-SimSiam encourage a small cosine similarity (about 0.3), which means that the representations are encouraged to be equivariant.
>
> > b) I would like to see an ablation on the “crop with size and scale” for the different prediction views. I understand that it is used to save memory; however, I wonder if it has an impact on performance. For example, in Fig. 3 the crop nicely centers on the object and removes background. Please conduct an experiment without the crop/resize, to make it feasible for the memory, just uniformly sample the rotation during training, i.e., an unbiased estimate of Eq. 2.
>
> Thank you for suggesting this ablation study. We added it to Appendix B. Below we summarize our results:
>
> CIFAR-10:
>
> Baseline: 92.0+-0.0
>
> E-SimCLR (small crop): 94.1+-0.0
>
> E-SimCLR (small crop, single rotation): 93.4+-0.0
>
> E-SimCLR (large crop, single rotation): 93.9+-0.0
>
> It seems like a single rotation with a large crop is almost as good as using 4 views and a small crop.
>
> > Maybe move Alg. 1 and Table 3 to the top of the page; I think this would make the paper neater.
>
> Thanks: please see the updated paper where we have incorporated your suggestion.
>
> > Might consider citing the following related works: Misra, Ishan, C. Lawrence Zitnick, and Martial Hebert. "Shuffle and learn: unsupervised learning using temporal order verification." Proc. ECCV. 2016; Lee, Hsin-Ying, et al. "Unsupervised representation learning by sorting sequences." Proc. ICCV. 2017; Mundhenk, T. Nathan, Daniel Ho, and Barry Y. Chen. "Improvements to context based self-supervised learning." Proc. CVPR, 2018.
>
> Thank you for the suggestion. We cited these works in the Related Work section.

---

### Official Review · Reviewer_tezr · 2021-11-02

**Correctness:** 4
**Technical Novelty And Significance:** 2
**Empirical Novelty And Significance:** 2
**Recommendation:** 6
**Confidence:** 2

**Main Review:**

* Paper is well written and easy to follow
* The proposed method is simple (adding an additional loss) and shows some encouraging results on different methods (SimCLR, SimSiam, and Barlow Twins) on two image datasets CIFAR-10 and ImageNet
- I find the results on ImageNet a little weak though, because it shows only results after 100 epoch of training. Longer training can change results (see https://arxiv.org/pdf/2011.10566.pdf - Table-4).
- The main weakness of the paper is that although it claims E-SSL as a more general method (which I would assume would replace I-SSL), the actual method boils down to adding an additional loss (which is not novel per se). Basically, the claim is stronger than the actual method/implementation.


**Summary Of The Paper:**

The paper advocates Equivariance Self-Supervised Learning (E-SSL) as a more general framework than the Invariance SSL (I-SSL), which is common in SoTA vision SSL methods, such as SimCLR and Barlow Twins. The paper starts with empirical evidence (Fig-1) that adding equivariance objectives (e.g. 4-fold rotation and vertical flips) to SimCLR can improve performance. The proposed E-SSL framework boils down to adding an additional equivariance objective (mainly 4-fold rotations) to popular I-SSL methods (Eqn. 1 & 2). The empirical results show encouraging results in CIFAR-10 and ImageNet. Finally, the paper applies the proposed method in two datasets of photonic data (regression task from 2D square periodic unit cells).

**Summary Of The Review:**

The paper claims proposing a general E-SSL framework, but ends up adding an additional loss from prior work (four-fold rotation prediction) to popular SSL methods.

---

> ### Author Response · Authors · 2021-11-17
> **1. Gains on ImageNet do not diminish; 2. Generality of E-SSL.**
>
> > I find the results on ImageNet a little weak though, because it shows only results after 100 epoch of training. Longer training can change results (see https://arxiv.org/pdf/2011.10566.pdf - Table-4).
>
> We agree that this might be a reasonable hypothesis for a large dataset such as ImageNet, but we do not observe that completely in our experiments. For example, Barlow Twins experiments show consistent 1% improvements with 200 and 300 epochs too. Naturally, CIFAR-10, CIFAR-100, Flowers-102 and PhC datasets benefit the most from E-SSL (relatively speaking), probably because there is less data, thus the inductive biases encouraged by E-SSL are underscored in the results. Please refer to the updated Table 3 in the revision.
>
> > The main weakness of the paper is that although it claims E-SSL as a more general method (which I would assume would replace I-SSL), the actual method boils down to adding an additional loss (which is not novel per se). Basically, the claim is stronger than the actual method/implementation.
>
> While we agree that our method has been demonstrated with the inclusion of an additional loss term, we actually believe that this is a flexibility worth appreciating about E-SSL, because we can encourage equivariance for the numerous I-SSL methods. We agree that in our paper we consider one simple way to encourage equivariance to representations, by predicting the transformation from the representation, as a proof-of-concept. Therefore, we hope to underscore our main claim in the study: that some transformations worsen existing SSL methods when encouraged to be invariant, and that when encouraged to be equivariant they improve SSL methods. To our best knowledge (see detailed comparison in the Related Work section) the complementarity nature of invariance and equivariance has not been explored in the literature (Figure 2), and in our paper we show that it can be neatly captured by the concept of E-SSL as a generalization to I-SSL.

---

> > ### Comment · Reviewer_tezr · 2021-12-01
> > **Updated scores**
> >
> > Thanks for your detailed response and updates of the paper. I have carefully read your updated sections, rebuttal, response to my review and other reviewers. My conclusion is that the paper is in a better shape to be accepted.

---

> ### Author Response · Authors · 2021-11-25
> **follow up**
>
> Dear Reviewer,
>
> We wonder if you’ve had a chance to read our rebuttal. Please, let us know if you have any remaining questions: we would love to hear them and respond in a timely manner.
>
> About the ImageNet results, we have added Table 3 in the revision which shows consistent gains with longer pre-training on ImageNet.
>
> We would like to clarify that our contribution is much more general than an addition of a four-fold rotation loss to popular SSL methods. Figure 1 in our paper shows the complementary nature of invariance and equivariance, where we consider many types of transformations. As Reviewer 9Qn8 observed, “The experiments in figure 1 lays out a clear roadmap for others to follow.” In our experiments with photonic crystals (Section 5 and Appendix E) we show that equivariance to four-fold translations and scaling (this experiment is a demonstration of E-SSL with a continuous group, inspired by Reviewer p4FW) improves I-SSL methods. Additionally, thanks to Reviewer 9Qn8 we showed the usefulness of equivariance to four-fold translations for the Flowers-102 dataset. All these losses for equivariance do not exist in prior works, to the best of our knowledge. Taken together, these experiments show the generality of our E-SSL framework and four-fold rotation is simply one specific implementation of our idea.
>
> Best wishes,
> The authors

---

### Official Review · Reviewer_9Qn8 · 2021-11-02

**Correctness:** 4
**Technical Novelty And Significance:** 3
**Empirical Novelty And Significance:** 3
**Recommendation:** 8
**Confidence:** 3

**Main Review:**

### Strengths:
-	I am familiar with idea of learning invariant representations and have seen it work in practice.  The idea to learn sensitivity using equivariance seems like logical and useful extension to me.   Moreover, I think the introduction is very nicely written and clearly explains the intuition of the idea.
-	The experiments in figure 1 lays out a clear roadmap for others to follow.  One may explore which transformations it may be beneficial to be insensitive or sensitive too.  There is a clear logical inverse correlation in which if it is hurts performance for features to be invariant to a transformation, it is likely to be beneficial to be sensitive to them.
-	The further experiments on CIFAR-10, ImageNet, and Photonic Crystals show a clear benefit to the method.  In particular, I like the combination of standard benchmark datasets and good baselines as well as interesting application domains where the proposed method may be truly be useful.

### Weakness/Limitations/Questions

-	The biggest question and issue for me is the following assumption: the distribution/dataset is assumed to contain only one element of each group orbit.  One can see this assumption is implicit in eqn. (2).  To the authors credit, it is spelled out explicitly in Proposition 1, but I think it is down-played a bit, when it should probably be explored more.  Taking the example of 90 degree rotations, the assumption is that every image has a correctly inferable orientation.  This is, in fact, what the loss of eqn. (2) is learning.  So if both a datapoint x and a datapoint rot(x) exist in the dataset, the model cannot learn eqn (2) correctly for both.
This assumption may hold for Imagenet which contains natural photos which have a right side up, but many types of computer vision tasks do not have such features: satellite photos, xrays, microscope images, astronomy images, etc.  In fact, domains in which orientation is arbitrary are exactly the domains in which equivariance has been most relevant in DL.  I believe the current implementation of eqn. (2) makes such domains inaccessible for the current method.

-	In particular, it seems the PhC dataset is of this kind (though I may not have understood correctly).  The authors say DOS is rotation invariant. Thus intuitively it seems an invariant representation would be better.  Thus it’s surprising to me a rotation sensitive representation works better.
It may be sensitivity to rotation only helps not due to the domain or task, but due to a bias in the creation of the dataset in which some implicit “correct orientation” is enforced.  Perhaps if the dataset contains many rotated views of each sample this issue would go away and some other form of representation learning would work better?  I’d welcome more explanation here.

-	The authors cite several related works which also learn features which are sensitive to translations.  How close is their method to these.  Do they also invoke equivariance? That is, I’m not sure whether equivariant semi-supervised learning (with faithful representations) is new to this paper or whether the contribution of this paper is the combination of equivariance to some transformations and invariance to others.  Please clarify if possible.

-	In table 3, it’s a bit hard for me to interpret the numbers.  Due to the normalization, values like 3.0 or 1.0 seem like very large errors and the difference between the methods seems rather small.   In that case, these results here are not that strong.  Do you have qualitative results?  I want to know it’s not the case that all models are simply failing at the task.


### Small Notes/Further Questions:
-	Mapping Sensative/Insensative onto the regular represention/trivial representation makes for an extreme dichotomy.  One might also consider representations between the regular and trivial rep.  In fact, this is exactly what the authors do in considering a quotient rep C_4/C_2 for the cyclic group in the phototonic-crystal experiment.
-	Can you clarify the ablated E-SSL models: “Linear enhancing predictor” and “No SSL augmentation in the enhanced views.”  I couldn’t find the description in the text.


**Summary Of The Paper:**

This work presents a framework for learning representations with invariances (insensitivity) to some transformations and sensitivities to others.  Previous work has only considered representations with insensitivity or sensitivity but not a combination.  The authors use the concept of invariance or equivariance to symmetry group transformations to learn such representations.  The advantage of the proposed method is demonstrated relative to SimCLR and SimSiam on Cifar-10, ImageNet, and a new scientific application domain: learning frequency responses of photonic crystals.  The authors also prove a theoretical characterization of when their methods works.

**Summary Of The Review:**

I quite like the idea and experiments, but I have some reservations related to my questions.  Depending on the response, I’d be happy to raise my score.

---

> ### Author Response · Authors · 2021-11-17
> **1. Important experiment on pre-training on a new dataset -- Flowers-102.**
>
> > The biggest question and issue for me is the following assumption: the distribution/dataset is assumed to contain only one element of each group orbit. One can see this assumption is implicit in eqn. (2). To the authors credit, it is spelled out explicitly in Proposition 1, but I think it is down-played a bit, when it should probably be explored more. Taking the example of 90 degree rotations, the assumption is that every image has a correctly inferable orientation. This is, in fact, what the loss of eqn. (2) is learning. So if both a datapoint x and a datapoint rot(x) exist in the dataset, the model cannot learn eqn (2) correctly for both.
> This assumption may hold for Imagenet which contains natural photos which have a right side up, but many types of computer vision tasks do not have such features: satellite photos, xrays, microscope images, astronomy images, etc. In fact, domains in which orientation is arbitrary are exactly the domains in which equivariance has been most relevant in DL. I believe the current implementation of eqn. (2) makes such domains inaccessible for the current method.
>
> We completely agree with you, and thank you for inviting a discussion about this assumption! Following the advice of Reviewer p4FW we have placed Section 5 earlier in the paper so that this assumption is clear from the beginning. To address your comment, note that our main observation in the paper is related to equation Eq 1, which we have made more general in the revised version. Eq. 2 is just an example of one way of doing E-SSL for ImageNet and CIFAR. To show complete generality, let’s take a dataset for which Eq. 2 may not work at first sight. This dataset is the Flowers-102 dataset (https://www.robots.ox.ac.uk/~vgg/data/flowers/102/index.html). In the spirit of Figure 1, let’s search for the useful transformations for this dataset. As you suggested, equivariance to rotation may not be useful, because invariance to rotation is a much more useful property. However, note that Flowers is just like the Blob PhC dataset! Roughly speaking, the dataset contains only one element of each group orbit of the four-fold translations group. Therefore, we conduct an experiment in which we encourage equivariance to four-fold translations, just like for the Blob PhC dataset in Appendix F. We observed 1-2% improvement in the kNN accuracy. We even observed benefit from encouraging equivariance to four-fold rotations, which is against the intuition that rotations should be invariant. This is probably due to the fact that some images in the dataset are not truly rotationally invariant (see examples of the data points in the paper). In short, while the assumption you pointed out (that every image has a biased orientation) may not hold for other domains, we are often able to find other transformations in those domains that E-SSL can provide benefits over the conventional contrastive learning (an example is the above-mentioned four-fold translations in the Flowers dataset) and the goal of our work is precisely to demonstrate that equivariance is a useful concept in the context of contrastive learning, and I-SSL in general.

---

> ### Author Response · Authors · 2021-11-17
> **2. Clarification on PhC and main contribution.**
>
> > In particular, it seems the PhC dataset is of this kind (though I may not have understood correctly). The authors say DOS is rotation invariant. Thus intuitively it seems an invariant representation would be better. Thus it’s surprising to me a rotation sensitive representation works better.
> It may be sensitivity to rotation only helps not due to the domain or task, but due to a bias in the creation of the dataset in which some implicit “correct orientation” is enforced. Perhaps if the dataset contains many rotated views of each sample this issue would go away and some other form of representation learning would work better? I’d welcome more explanation here.
>
> Yes indeed your intuition is correct. In the PhC problem, the rotation invariance is governed by the physics of the problem and we would intuitively think that an invariant representation is better. The same holds for other transformations governed by the physics, like translation and scaling invariances (this is a new experiment to show E-SSL also works for continuous groups following the suggestion of Reviewer p4FW; please see Appendix E). In our paper, we show two datasets where we deliberately invoke biases during the creation of the dataset; in the Blob dataset, features are centered and we later increase sensitivity to translations and in the Gpm dataset, input unit cells have a horizontal mirror symmetry and we later increase sensitivity to rotations. For both datasets, we show that 1) adding the ‘biased’ transformation to SimCLR worsens performance and 2) increasing sensitivity to the ‘biased’ transformations via E-SimCLR improves performance. In short, our experiments show that, depending on the domain or dataset, naively encouraging all invariances that are a-priori known from the physics of the problem is not always useful (and can be harmful) and in these domains, the concept of equivariance can be used to improve performance.
>
> > The authors cite several related works which also learn features which are sensitive to translations. How close is their method to these. Do they also invoke equivariance? That is, I’m not sure whether equivariant semi-supervised learning (with faithful representations) is new to this paper or whether the contribution of this paper is the combination of equivariance to some transformations and invariance to others. Please clarify if possible.
>
> The contribution of this paper is the combination of equivariance to some transformations and invariance to others. As discussed in the Related Work section and in Figure 2, other papers invoke equivariant semi-supervised learning with faithful representations only, i.e. they neglect the invariance (trivial representations) part. We clarified that part in the introduction by updating the bullet points with contributions and distinguishing our work in the Related Work section.
>
> > In table 3, it’s a bit hard for me to interpret the numbers. Due to the normalization, values like 3.0 or 1.0 seem like very large errors and the difference between the methods seems rather small. In that case, these results here are not that strong. Do you have qualitative results? I want to know it’s not the case that all models are simply failing at the task.
>
> The values in Table 3 are the relative errors between the predicted and true DOS spectrum indicated in %. An error of 1-3% is rather reasonable for this problem and is not an indication of catastrophic failing. To support this statement, we refer you to Figure 4b of the paper, https://arxiv.org/pdf/2110.08406.pdf, which shows what various errors look like when using the same metric. Indeed we agree that the differences between the methods may seem rather small; however we note that they are mostly statistically significant (as indicated by the error bars). One reason for the small difference between the methods for the PhC problem is because, unlike image classification, the SimCLR objective is not immediately useful for the DOS prediction task which involves predicting a high dimensional spectrum. The (E-)SimCLR step was used simply as a mechanism to invoke known invariances (or equivariances) of the input unit cells. Due to the sophistication in the labels involved, we need to fine-tune a dense non-linear DOS predictor network (compared to a linear layer for image classification) when using label information and hence the final performance is largely weighed upon by the fine-tuning step and this cushions the improvements we get from the pre-training stage. A possible way to overcome this is to also include weak label information in the pre-training stage as explored in https://arxiv.org/pdf/2110.08406.pdf; however we will leave this for future work.

---

> ### Author Response · Authors · 2021-11-17
> **3. Clarification on representation theory of groups and ablated E-SSL experiments.**
>
> > Mapping Sensative/Insensative onto the regular represention/trivial representation makes for an extreme dichotomy. One might also consider representations between the regular and trivial rep. In fact, this is exactly what the authors do in considering a quotient rep C_4/C_2 for the cyclic group in the phototonic-crystal experiment.
>
> Thank you for the suggestion! We wanted to have an introduction that is representation-theoretic-friendly :), that’s why we focused on insensitive/ sensitive. We are really happy that we speak the same language! We clarified your observation in the first paragraph in Section 3 in the revision.
>
> > Can you clarify the ablated E-SSL models: “Linear enhancing predictor” and “No SSL augmentation in the enhanced views.” I couldn’t find the description in the text.
>
> Thank you for catching this. We mean “Linear predictor for equivariance” and “No SSL augmentation in equivariance views”. The former experiment shows that a linear predictor is less optimal than a non-linear one, and the latter experiment shows that typical SSL augmentation (i.e. the standard SimCLR augmentation strategy) is useful for the prediction objective. The latter point is also consistent with the experiment in Appendix B. To our knowledge, we are the first in the literature to observe the importance of using augmentation in contrastive learning for the rotation prediction task. We updated the paper accordingly.

---

> ### Author Response · Authors · 2021-11-25
> **follow up**
>
> Dear Reviewer,
>
> We wonder if you've had a chance to read our rebuttal. Please, let us know if you have any remaining questions: we would love to hear them and respond in a timely manner.
>
> Best wishes,
> The authors

---

> > ### Comment · Reviewer_9Qn8 · 2021-11-29
> > **Thank you**
> >
> > I have read the revised paper, other reviews, and author responses.  I appreciate the authors answers, revisions, and additional experiments.  I think the extension to an infinite group adds some variety to the paper and shows the flexibility of the method.
> >
> > I'm less clear on the impact of the flower dataset.  It seems to reaffirm the finding that encouraging equivariance is about the importance of the data having a single element in each group orbit.  In this case, it seems natural images of flowers have canonical orientations.  If it is important to pay attention to the orientation of the image for classifying flowers, is that because some aspect of orientation is inherently important to the task or is it just that the same bias affects the test and train set and can thus be used as shortcut?
> >
> > Overall, I really like the proposed conceptual framework and I think the paper makes a worthwhile contribution.  Though I remain unsure about limitations of the method, I think it provides enough benefit as it stands, and can hopefully inspire future work to address those limitations.
> >
> > For example, what if the class label depends on the orientation (say, classifying road signs including left and right arrows), in this case sensitivity to the orientation is paramount, yet the current method for encouraging sensitivity would fail since both $x$ and $gx$ exist in the data.

---

> > > ### Author Response · Authors · 2021-11-30
> > > **Addressing the case when $x$ and $gx$ exist in the data**
> > >
> > > > I'm less clear on the impact of the flower dataset. It seems to reaffirm the finding that encouraging equivariance is about the importance of the data having a single element in each group orbit. In this case, it seems natural images of flowers have canonical orientations. If it is important to pay attention to the orientation of the image for classifying flowers, is that because some aspect of orientation is inherently important to the task or is it just that the same bias affects the test and train set and can thus be used as shortcut?
> > >
> > > Intuitively, the latter is more important for Flowers-102 classification. We argue that if the shared bias exists, then we should exploit it. With this dataset we showed we can usually find such shared biases and use E-SSL to take advantage of them. However, of course this depends on the downstream task, of which you give a nice example below. We have follow up experiments to address your suggestion. Please, see them below.
> > >
> > > > For example, what if the class label depends on the orientation (say, classifying road signs including left and right arrows), in this case sensitivity to the orientation is paramount, yet the current method for encouraging sensitivity would fail since both $x$ and $gx$ exist in the data.
> > >
> > > This is a nice example of a dataset, which invites a natural generalization of our method in the spirit of [1]. If $x$ and $gx$ exist in the data then we can modify E-SSL minimally to form a useful objective. In particular, we can set the objective for $p_2$ to predict the relative orientation between two data points, i.e. given $x$, we form $g'x$ by sampling $g'$ from $G$ uniformly, and then predict $g'$ from $p_2(z)$, where $z = [f(x), f(g'x)]$ is the concatenation of the two representations. This modification requires minimal change in our framework and can address the types of data you propose. Most importantly, because it is doing relative prediction, E-SSL will not fail on your example.
> > >
> > > To test the case when $x$ and $gx$ exist in the data, we can artificially modify CIFAR-10 so that any rotation of an image can appear in the dataset. To follow your example closely, we consider the downstream task of predicting the rotation orientation of an image. Our hypothesis is that the modified E-SimCLR will be better than SimCLR, which is what we observe in our results below. Intuitively, SimCLR is not able to capture the orientation of the image, while the modified E-SimCLR is able to, because the latter predicts relative orientation.
> > >
> > > The experimental setting is as follows: we pretrain for 100 epochs. The predictor for equivariance’s input dimension is doubled, because we concatenate two representations.  We set $\lambda$ to 0.4. All other hyperparameters are the same as the rest of the CIFAR-10 experiments.
> > >
> > >
> > > ===
> > >
> > > SimCLR (baseline) on unbiased CIFAR-10 dataset, 4-way rotation orientation classification linear probe (%): 67.1+-0.1
> > >
> > > E-SimCLR (ours) on unbiased CIFAR-10 dataset, 4-way rotation orientation classification linear probe (%): **71.2+-0.1**
> > >
> > > ===
> > >
> > > The scenario you described is also highly relevant in other domains such as in Photonic Crystals. For example, we can modify the Photonic Crystals setup and remove the "orientation bias" of the datasets while predicting a different property, the band structures. Unlike the DOS, the PhC band structures are not invariant to rotations and so we can consider the group of four-fold rotations. This setup would fit the scenario you described since 1) both $x$ and $gx$ exist in the data due to the lack of bias and 2) the downstream task is sensitive to rotation. We will explore the above framework on this setup in future work.
> > >
> > > We will report on these preliminary results in the appendix and will leave this direction for future work.
> > >
> > > [1] Agrawal, Pulkit, Joao Carreira, and Jitendra Malik. "Learning to see by moving." Proceedings of the IEEE international conference on computer vision. 2015.

---

> > > > ### Comment · Reviewer_9Qn8 · 2021-11-30
> > > > **Thank you for this modified version and additional experiments**
> > > >
> > > > Thank you for this response.  Your modified version of $p_2$ directly addresses my concern with the method.   I also appreciate that the experiment you performed matches my proposed scenario and demonstrates that this modified relative version of the task also works.  I will increase my score.

---

### Official Review · Reviewer_7QWR · 2021-11-02

**Correctness:** 3
**Technical Novelty And Significance:** 2
**Empirical Novelty And Significance:** 2
**Recommendation:** 6
**Confidence:** 4

**Main Review:**

The idea is simple and straightforward, making it easy to understand. The idea makes sense to me and the experiments show good results.

It is not clear how the authors did in the introduction to encourage the model to be sensitive or not sensitive to a specific transformation. I would imagine that for encouraging insensitive, the authors did not do anything specific, and for encouraging it to be sensitive, the authors used an additional branch to predict just like the main framework. Still, it would be great if the authors could make it more clear.

I do not really buy the equivariance story. The story and the title are somewhat overclaiming. The authors bring the definition of equivariance in the introduction; however, can the authors be more specific what is the $T'_g$ in their framework? I do not see any post transformation on top of the model output on the newly added branch but just labels. The model did not really learn equivariance but used some auxiliary pretext tasks to help learn better representations.

It would be great to see how the model performs with longer training iteration on ImageNet-1K, to confirm the gain will not be diminished.

**Summary Of The Paper:**

This paper presents a simple solution to improve the existing self-supervised representation learning by adding an additional branch to predict the rotations. Experiments are conducted on CIFAR-10, ImageNet, and two PhC datasets and the results show that the model with the additional branch can outperform the corresponding baselines.

**Summary Of The Review:**

Overall I think this paper has its own contribution. It finds that using an additional pretext header can improve the performance. However, I don't think the story itself is convincing enough and believe that might mislead readers. How the model performs with longer training iteration on ImageNet-1K also remains unclear. These prevent me from giving a higher rating.

--post rebuttal

After reading the authors response I raise my rating to 6.

---

> ### Author Response · Authors · 2021-11-17
> **1. Clarifying the E-SSL framework; 2. Gains on ImageNet do not diminish.**
>
> > It is not clear how the authors did in the introduction to encourage the model to be sensitive or not sensitive to a specific transformation. I would imagine that for encouraging insensitive, the authors did not do anything specific, and for encouraging it to be sensitive, the authors used an additional branch to predict just like the main framework. Still, it would be great if the authors could make it more clear.
>
> Thank you for allowing us to clarify! Actually, encouraging insensitivity is by adding the transformation to the SimCLR data augmentation. We updated the paper accordingly with the following addition: “We encourage insensitivity by adding the transformation to the SimCLR data augmentation, and sensitivity by predicting it (see Section Experiments).”
>
> We hope that this clarification underscores the main observation of our study: that some transformations worsen existing SSL methods when encouraged to be insensitive, and that when encouraged to be sensitive they improve SSL methods. To our best knowledge (see detailed comparison in the Related Work section) the complementary nature of insensitivity/ sensitivity has not been explored in the literature, and in our paper we show that it can be neatly captured by the concept of equivariance as a generalization to I-SSL.
>
> Finally, note that we explore a variety of transformations in Figure 1. As we mention in our Discussion section “While we see combinations of transformations as promising future work, we focused on encouraging sensitivity to a single transformation to make a clear presentation of the E-SSL framework.” Additionally, as Reviewer 9Qn8 observes, “The experiments in figure 1 lays out a clear roadmap for others to follow.”
>
> > I do not really buy the equivariance story. The story and the title are somewhat overclaiming. The authors bring the definition of equivariance in the introduction; however, can the authors be more specific what is the $T_g'$
>  in their framework? I do not see any post transformation on top of the model output on the newly added branch but just labels. The model did not really learn equivariance but used some auxiliary pretext tasks to help learn better representations.
>
> Note that the concept of equivariance is a useful concept to capture the complementarity of the insensitivity/ sensitivity to transformations, as mentioned above. $T_g’$ is not a component we explicitly control in our framework. However, thinking about the possible choices of $T_g’$ is useful to understand the importance of complementarity of transformations. $T_g’$ is implicitly trained to be the identity in prior frameworks (SimCLR, SimSiam, Barlow Twins, etc.), while we implicitly discourage it to be the identity. To provide more details, we refer you to the mathematical formalism in the Introduction, Section 5.2 (Section 3 in the revision), the discussion around Proposition 1, as well as the last paragraph of the Related Work section.
>
> Additionally, thanks to Reviewer p4FW we have conducted a measurement analysis in Appendix C.3 that shows that our objective indeed encourages equivariance in the representations.
>
> > It would be great to see how the model performs with longer training iteration on ImageNet-1K, to confirm the gain will not be diminished.
>
> We confirmed that the gains do not diminish in the added table (Table 3 in the revision).

---

> > ### Comment · Reviewer_9Qn8 · 2021-11-29
> > **T_g'**
> >
> > While $T_g'$ is unclear for $f$, I suppose for $p_2 \circ f$, $T_g$ is image rotation and $T_g'$ is cyclic shifting through $\lbrace 0, 90, 180, 270 \rbrace$.

---

### Author Response · Authors · 2021-11-17
**General response to Reviewers**

We would like to thank the Reviewers for their constructive suggestions, which allowed us to improve our paper! We believe that we were able to respond in depth to all of the concerns, and kindly ask the Reviewers to consider increasing their scores. We updated the paper accordingly (please see additions in blue). Below we summarize our revision.

# Experiments

For Reviewers “7QWR” and “tezr” who wanted to see more experiments on ImageNet, we addressed their comments with the added Table 3 in the revision. The intuition of the Reviewers was that with more pretraining the gains from E-SSL will diminish. We agree that this might be a reasonable hypothesis for a large dataset such as ImageNet, but we do not observe that completely in our experiments. For example, Barlow Twins experiments show consistent 1% improvements with 200 and 300 epochs too. Naturally, CIFAR-10, CIFAR-100, Flowers-102 and PhC datasets benefit the most from E-SSL (relatively speaking), probably because there is less data, thus the inductive biases encouraged by E-SSL are underscored in the results.

# Novelty

We hope to underscore our main claim in the study: that some transformations worsen existing SSL methods when encouraged to be invariant, and that when encouraged to be equivariant they improve SSL methods. To our best knowledge (see detailed comparison in the Related Work section) the complementarity nature of invariance and equivariance has not been explored in the literature (see Figure 2 and Related Work Section), and in our paper we show that it can be neatly captured by the concept of E-SSL as a generalization to I-SSL.

We would like to emphasize that E-SSL is a more general framework that works for datasets beyond CIFAR and ImageNet with rotation prediction. We demonstrated this in Figure 1 for vertical flips, 2x2 jigsaws, four fold gaussian blurs and color inversion. We demonstrated this in photonics science too where we found that in some settings equivariance to four-fold rotations is useful, and in others -- equivariance to four-fold translations. Finally, thanks to Reviewer 9Qn8 we demonstrated the usefulness of E-SSL with four-fold translations *and*, surprisingly, even four-fold rotations, on the Flowers-102 Dataset.

Why did we focus on rotation for CIFAR-10 and ImageNet, even though rotation alone has been studied in the literature? Even though it is a simple idea, it is not trivial to perfect the implementation: there are a lot of tricks to get the best performance. All of the Reviewers agreed that the idea is sound. Please, keep in mind that Figure 3 and Algorithm 1 focus on rotation prediction precisely because it exists in the related work. Thus, it serves as a good introduction for the reader. Finally, as Reviewer p4FW kindly appreciated, we will release our code, so that people can start experimenting with these tricks.

More importantly, Figure 1 shows that our method goes beyond existing work, because it shows we study many other transformations. So the idea is general, but as a proof-of-concept we focused on rotation for getting our best results on ImageNet and CIFAR-10. Note that we show the idea for PhC, where we use four-fold translations as well (four-fold translations do not appear in related work). And we showed it on Flowers-102 too, thanks to Reviewer 9Qn8. Thanks to Reviewer p4FW we also showed that E-SSL works for infinite groups (see Appendix E).

Finally, we would also like to note that the concept of Equivariant SSL can be considered in NLP as well. We were interested to find the work of Meng et al. [https://arxiv.org/pdf/2102.08473.pdf] that combines contrastive pretraining (invariance) with text corruption prediction (equivariance).

## Additional experiments beyond what the Reviewers suggested (added to Appendix C.2):

We show that the same ResNet18-setting works for CIFAR100 too (notice the sizable 3-4% improvement):

SimCLR:

CIFAR100 baseline: 65.8+-0.0

CIFAR100 E-SSL: 69.5+-0.1

SimSiam:

CIFAR100 baseline: 65.8+-0.1

CIFAR100 E-SSL: 69.3+-0.1

We show that it works for fully connected backbones on CIFAR-10 as well (3*32*32->2048->2048->512 with 1D BNs and ReLUs), optimization setting is the same:

SimCLR:

CIFAR10 baseline: 70.5+-0.0

CIFAR100 E-SSL: 73.8+-0.1

SimSiam:

CIFAR10 baseline: 70.9+-0.0

CIFAR10 E-SSL:73.5+-0.1

---

### Public Comment · ~Yifei_Wang1 · 2022-02-07
**A reminder on a related paper**

Congratulations on this inspiring paper being accepted! We would like to point out our NeurIPS 2021 paper, [Residual Relaxation for Multi-view Representation Learning](https://arxiv.org/pdf/2110.15348.pdf). As in this work, we also combine invariance-based SSL (e.g. BYOL, SimSiam) with equivariance-based SSL (e.g., Rotation) and demonstrate its benefits. It is exciting to see this idea has also been studied from different perspectives and worked under a different solution. It would be nice to discuss it in the final version.

---

> ### Public Comment · ~Rumen_Dangovski1 · 2022-02-08
> **Thank you for pointing out your insightful work!**
>
> We will discuss your paper in the Related Work section in the Camera Ready Revision. We believe that both our E-SSL and your Residual Relaxation approaches are effective ways to “bridge the two existing methodologies for representation learning,” as you suggest in your paper.

---

### Public Comment · ~Rumen_Dangovski1 · 2022-03-16
**Camera Ready Revision Posted**

We would like to thank the anonymous reviewers, area chairs and program chairs for helping us make our paper stronger. We also obtained a 72.5% linear probe with E-SimCLR after 800 epochs of ImageNet-1K pre-training. We included all additional experiments in the camera ready revision.

All of our minimal code and instructions for reproducing a wide range of strong SSL baselines and their E-SSL improvements, new datasets of photonics crystals that can aid new SSL research, and pre-trained ImageNet-1K models, are available at https://github.com/rdangovs/essl. We believe that this repository would be of significant value to the community and we plan to maintain it. It is currently being used for follow-up research in SSL.

---

### Decision · Program_Chairs · 2022-01-20

**Decision:**

Accept (Poster)

**Comment:**

This paper proposes an extra loss to add on top of the contrastive learning. The contrastive learning seek representations invariant to transformation, while the extra loss the authors proposed encourage representations to be equivariant to the transformation (i.e. retain information about transformation in later representations). While reviewers and I agree this is a sensible motivation, and acknowledge good results that authors have obtained, the fact that most, if not all, improvement is combing from the 4-way rotation transformation is a bit unsatisfactory. Furthermore, this additional loss was proposed before and is actually quite well known, so the actual novelty in the proposed technique is somewhat limited. Nevertheless, this paper provides a comprehensive evaluation, obtaining a reasonable improvement, and makes a good case for using an equivariant seeking loss. The authors are strongly encouraged to release their code (including training details for reproducing ImageNet results) as the improvements they present are central to the acceptance.